# Capacitive piezotronics

Luying Xu[1,2,3,6], Zhuangzhuang Zhang[1,2,3,6], Gaobo Wang[1,2,3], Yixuan He[1,2,3], Junyi Zhai [1,2,3], Weiguo Hu [1,2,3], Libo Chen [4], Longfei Wang [1,2,3] ✉, Shuhai Liu [5] ✉ & Zhong Lin Wang [1,2] ✉

Interface engineering by polarization derives a plethora of distinctive phenomena. Most of them focus on modulation of barrier height for controlling carrier transport of direct-current electronics. However, modulating interface width under alternating-current settings and its resultant effects have not been explored. Here, we report the capacitive piezotronics, which utilizes piezoelectric polarization to control the interface width of heterostructures and modulate junction capacitance at high frequency. The built-in electric potential and the interface width can be reversibly tuned with amplitude as high as 0.11 V and 10.5 nm, which presents a high strain sensitivity ( > 110 fF/mbar), and surpasses that of commercial capacitive pressure sensors ( ~ 0.1-0.7 fF/mbar). It possesses a capacity of mechanically tuning transmission signal of communication systems with an amplitude > 11 kHz, and substantially improving the filtering characteristics particularly for high frequency noise ( > 300 kHz). The strain-tuned alternating-current electronics offer a distinctive approach for high quality communication.

Interfaces of materials and heterostructures lie at the heart of the laws in condensed-matter physics[1]. Manipulation of interfaces allows emergent effects and functionalities[2], which is critical for designing the post-Moore's devices. Recently, interface engineering, which utilizes piezoelectric[3–6], pyroelectric[7,8], ferroelectric[9], or flexoelectric[10–12] effect to induce polarization in semiconductors, has inspired emerging fields such as resistive piezotronic effect[13–15], pyro-phototronic effect[16–18], ferroelectronic effect[19–21], flexoelectronic effect[22–24] and flexo-photovoltaic effect[25,26], making great progress towards next-generation electronics/optoelectronics, advanced communications and artificial intelligence. It provides an ideal platform for exploring the fundamental couplings between polarization and intriguing physical processes[27]. Among these couplings, the resistive piezotronic effect is a typical and important phenomenon of interface engineering by piezoelectric polarization[28]. Its core concept is that the strain-induced piezoelectric polarization at semiconductor interface can effectively regulate the barrier height, so as to control the carrier transport of direct-current (DC) electronics, resulting in a substantial

change of resistance (Fig. 1a). As an inherent effect in non-centrosymmetric semiconductors, this resistive piezotronic effect has made a profound impact on the design and fabrication of DC electronics for mechanosensation[29,30], human-machine interfacing[31,32] and robotics[33–35]. The key to achieving the resistive piezotronic effect is to tune the interface barrier height by using piezoelectric polarization, which has also been the focus of interface engineering by polarization in recent years.

In addition, the interface width of heterostructures, referring to the width of the region where the potential energy is higher than that in the surrounding space, also plays a critical role in semiconductor devices[2,36], especially those regarding communication applications such as high-frequency transport[37–39] and high-cutoff-frequency radio-frequency diode[40,41]. It is closely associated with the distribution of built-in electric field and the band structure near the interface, which controls the processes of charge carrier separation, transport, relaxation or recombination within the interface region[1]. Actually, key properties such as junction capacitance and carrier dynamics under

[1]Beijing Institute of Nanoenergy and Nanosystems, Chinese Academy of Sciences, Beijing, China. [2]School of Nanoscience and Engineering, University of Chinese Academy of Sciences, Beijing, China. [3]Beijing Huairou Laboratory, Beijing, China. [4]Division of Solid-State Electronics, Department of Electrical Engineering, Uppsala University, Uppsala, Sweden. [5]Institute of Nanoscience and Nanotechnology, School of Materials and Energy, Lanzhou University, Lanzhou, Gansu, China. [6]These authors contributed equally: Luying Xu, Zhuangzhuang Zhang. ✉e-mail: lfwang12@binn.cas.cn; liushuhai1991@live.cn; zhong.wang@mse.gatech.edu

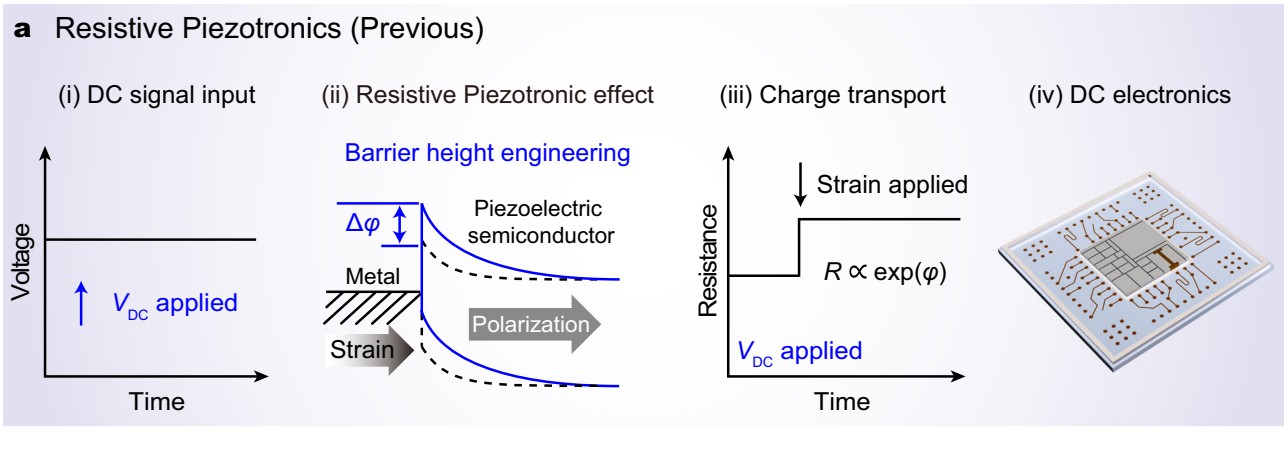

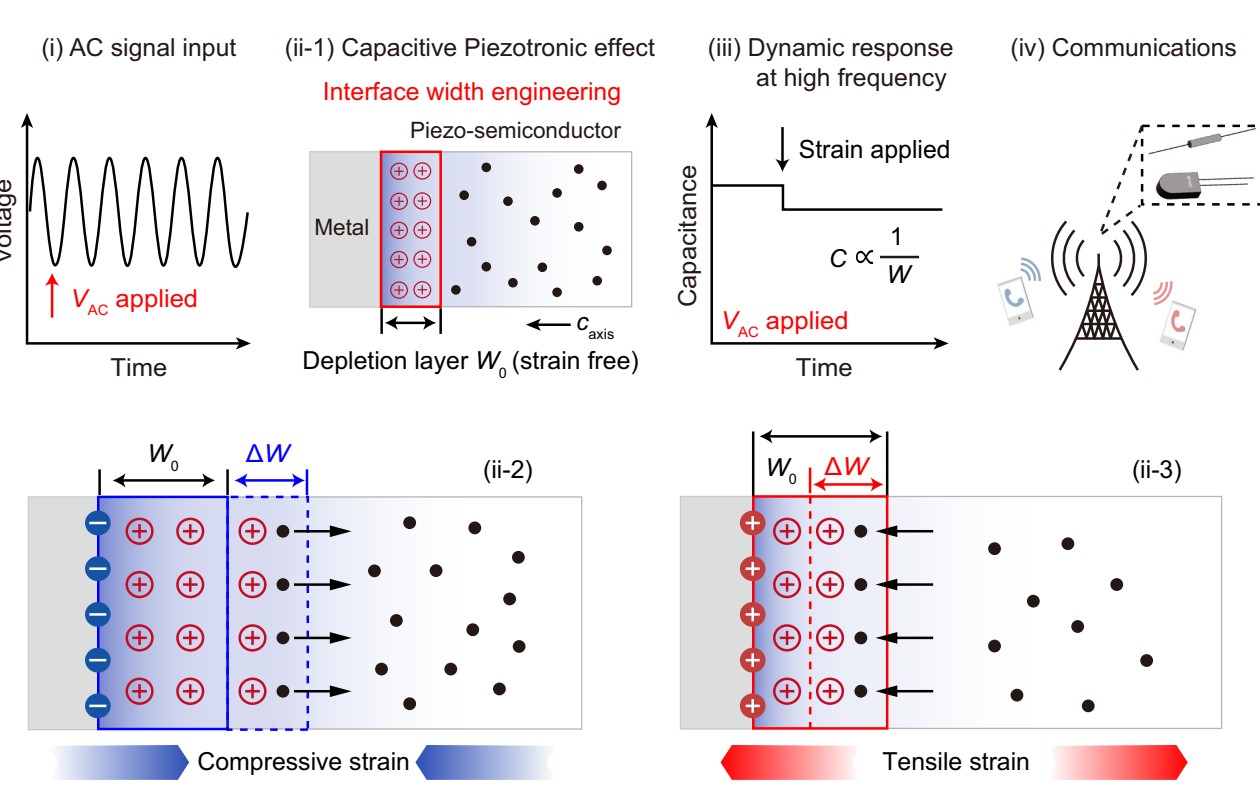

**Fig. 1 | Mechanism of the capacitive piezotronics. a** Concept of previous resistive piezotronics. With a direct-current (DC) signal $V_{DC}$ input (i), the strain-induced piezoelectric polarization at the Schottky contact can effectively regulate the barrier height $\varphi$ (ii), resulting in a substantial change of resistance (iii). The resistive piezotronic effect has made a profound impact on the design and fabrication of DC electronics (iv). **b** Concept of the capacitive piezotronics. With alternating-current (AC) signal $V_{AC}$ input (i), the interface width $W$ of the Schottky junction can be precisely controlled via capacitive piezotronic effect (ii), enabling a dynamic regulation of junction capacitance (iii). The capacitive piezotronics provides a strategy for mechanically tunable AC electronics in high-quality communication (iv). (ii-1), (ii-2), and (ii-3) illustrate the detailed modulation processes of the capacitive piezotronic effect under free, compressive and tensile strain, respectively. $W_0$ is the initial interface width of the Schottky junction, $W = W_0 + \Delta W$.

high-frequency alternating-current (AC) settings can be effectively controlled through modulating the interface width[2], providing a possible scheme for advanced performance of AC electronics/optoelectronics. Particularly, with the wide use of sub-10 nm (even sub-5 nm) materials and the development of device miniaturization in present-day technology, the interface width becomes more and more important to device performance. Generally, it can be designed by traditional methods such as atomic-level construction with precision growth equipment, material optimization (with different work functions or dielectric properties), elemental doping, etc., but keeps completely fixed and cannot be modulated further once it is constructed. Some preliminary attempts have been made to modulate the interface width using external stimuli, such as mechanical or optical stimuli[42]. However, most of these prior works have not provided systematic experiments and clear mechanism explanations. Thus, achieving dynamic manipulation of the interface width without changing the constructed device structure, and exploring the resultant effect are urgently needed, especially for AC electronics/optoelectronics.

Here, we proposed the capacitive piezotronics, demonstrating that polarization potential in piezoelectric semiconductor can be used as a gate signal to reversibly tune the interface width of heterostructures, so as to control the junction capacitance under high frequency AC settings ( ~ kHz-MHz). A substantial capacitive piezotronic effect was discovered in both single and dual Schottky junctions, in which the interface width can be reversibly tuned as high as 10.5 nm and 5.1 nm, respectively. It demonstrates a high sensitivity ($\Delta C/p > 110$ fF/mbar in a single Schottky junction) in tuning the junction capacitance, largely outperforming that of commercial MEMS-based capacitive pressure sensors ( ~ 0.1–0.7 fF/mbar[43,44]). Specifically, it can also be to mechanically tune the oscillation frequency of a typical resonant circuit with a frequency shift > 11 kHz, and substantially increase the cutoff frequency of a filtering circuit. This work provides an in-depth understanding of the discovered capacitive piezotronics towards next-generation AC electronics, demonstrating their huge potential in high-quality communication.

## Results

### Mechanism of the capacitive piezotronics

Figure 1b illustrates the main principle of the capacitive piezotronics. The amount of charged donors or acceptors within the depletion region can be modulated by an external voltage. Under AC settings, the applied AC voltage ($V_{AC}$) results in a periodic migration of free carriers at the interface, thereby inducing a fluctuation in the amount of charged atoms within the depletion region. This dynamic behavior resembles a charge-discharge process in a conventional capacitor. Notably, a charge-discharge process has a typical time constant, finally resulting in a substantial phase shift. Thus, a Schottky junction manifests a measurable capacitive characteristic under AC settings, and this capacitance is directly governed by the amount of charged atoms within the depletion region.

In a typical Schottky junction, the interface width ($W$) specifically refers to the width of the depletion layer at the Schottky interface, where the potential energy is higher than that in the surrounding space. The interface width ($W$) is inversely proportional to the junction capacitance ($C_j$)[2]:

$$\frac{C_j}{S} = \frac{\varepsilon_s}{W}, \tag{1}$$

where $S$ is the area of the interface and $\varepsilon_S$ represents the dielectric constant of the semiconductor. On this basis, the interface width could act as the key factor for the capacitive piezotronics.

When a strain or stress is applied on a piezoelectric semiconductor in a Schottky junction, a piezoelectric polarization field is generated at the Schottky interface due to the piezoelectric effect, which significantly influences the migration and redistribution of carriers at the interface, so as to change the interface width (Fig. 1b(ii))[45]. Taking a metal/$n$-semiconductor Schottky junction as an example, the negative piezoelectric polarization charges induced by compressive strain will repel electrons away from the interface, resulting in an increase in the amount of the charged donor atoms in the depletion layer. Thus, the interface width increases while the junction capacitance decreases (Fig. 1b(ii-2)). Conversely, the tensile strain/stress-induced positive piezoelectric polarization charges attract electrons near the interface, leading to a decrease in the amount of the charged donor atoms and the interface width, and an increase in the junction capacitance (Fig. 1b(ii-3)). Thus, the strain-induced piezoelectric polarization can effectively modulate the interface width, so as to control the junction capacitance under AC settings. This is the capacitive piezotronic effect, which differs from the previous resistive piezotronic effect that uses piezopotential to control the charge carrier transport of DC electronic devices. The use of inner-crystal piezoelectric polarization potential as a controlling signal, to modify the

interface width and achieve precise control of junction capacitance under AC settings, especially at high-frequency systems, is the basis of capacitive piezotronics. The modulation mechanism largely extends the realm of piezotronics to AC applications from a distinctive perspective, and provides a strategy for developing mechanical-stimulation-controlled devices in high-frequency communications (Fig. 1b(iv)).

### Relationship between capacitive piezotronics, resistive piezotronics and traditional capacitive effect

The capacitive piezotronics is an extension of the previous resistive piezotronics, but shows significant divergence in both basic principle and application scope. The fundamental nature of capacitive and resistive piezotronics is both an interface effects, which asymmetrically modulate local contacts at different terminals of a device using piezoelectric polarization. This piezotronic modification on the terminals can modify its electrical properties, thereby changing the impedance ($Z$) of the whole device, which can be described as:

$$Z = R + X = R + i\left(\omega L - \frac{1}{\omega C}\right), \tag{2}$$

where $R$, $X$, $L$ and $C$ represent resistance, reactance, inductance and capacitance of the device, respectively, $\omega$ is the angular frequency of AC signal, and $i$ is the imaginary unit. The previous resistive piezotronic effect is primarily characterized under DC conditions ($\omega = 0$), thus the piezotronic modification is reflected by the resistance ($R$) of the semiconductor device, which is closely related to its interface barrier height. In contrast, the capacitive piezotronics actually aims at studying the piezotronic modification on semiconductor devices under AC conditions ($\omega > 0$), where the reactance ($X$), particularly its capacitive component ($C$), becomes a critical parameter. In semiconductor heterostructures, such as $p$-$n$ junctions, Schottky junctions and tunneling junctions, this capacitive component is intrinsically associated with the junction capacitances, which are determined by the corresponding interface widths. Thus, the capacitive piezotronic effect focuses primarily on the interfacial electric properties under AC conditions, distinguishing it from the resistive piezotronic effect. As inherent effects in non-centrosymmetric semiconductors, the resistive piezotronics have made a profound impact on the design and fabrication of DC electronics for mechanosensation, while the capacitive piezotronics show exceptional prospects in AC electronics, such as high-frequency transport and communications.

Furthermore, the capacitive piezotronics significantly differ from but still have some relation with the traditional capacitive effect (Supplementary Note 7). Although both of these two effects ultimately modulate the capacitance of the device, the mechanism is completely different. The capacitive effect realizes variation in capacitance by changing the gap between two electrodes, which is a volume effect that regulates the capacitance without polarity. While the capacitive piezotronics uses piezoelectric polarization to asymmetrically modulate the interface width at two termini of Schottky junctions, which is an interface effect that enables substantial modulation of junction capacitance in semiconductor devices. The sensor, based on the capacitive piezotronic effect, is therefore a distinctive kind of capacitive device, which is well-suited for micro-nano structural integration, with high sensitivity and much miniaturization.

### Capacitive piezotronics in a single Schottky junction

**Theoretical simulations of the capacitive piezotronic effect.** To demonstrate the principle of the capacitive piezotronic effect, we performed Finite Element Analysis (FEA) on a Schottky junction based on Ga-polar (0001) $n$-GaN at zero bias, where the intrinsic piezoelectric $c$-axis of $n$-GaN is oriented perpendicular to the interface (Method). As the compressive strain increases, negative piezoelectric polarization

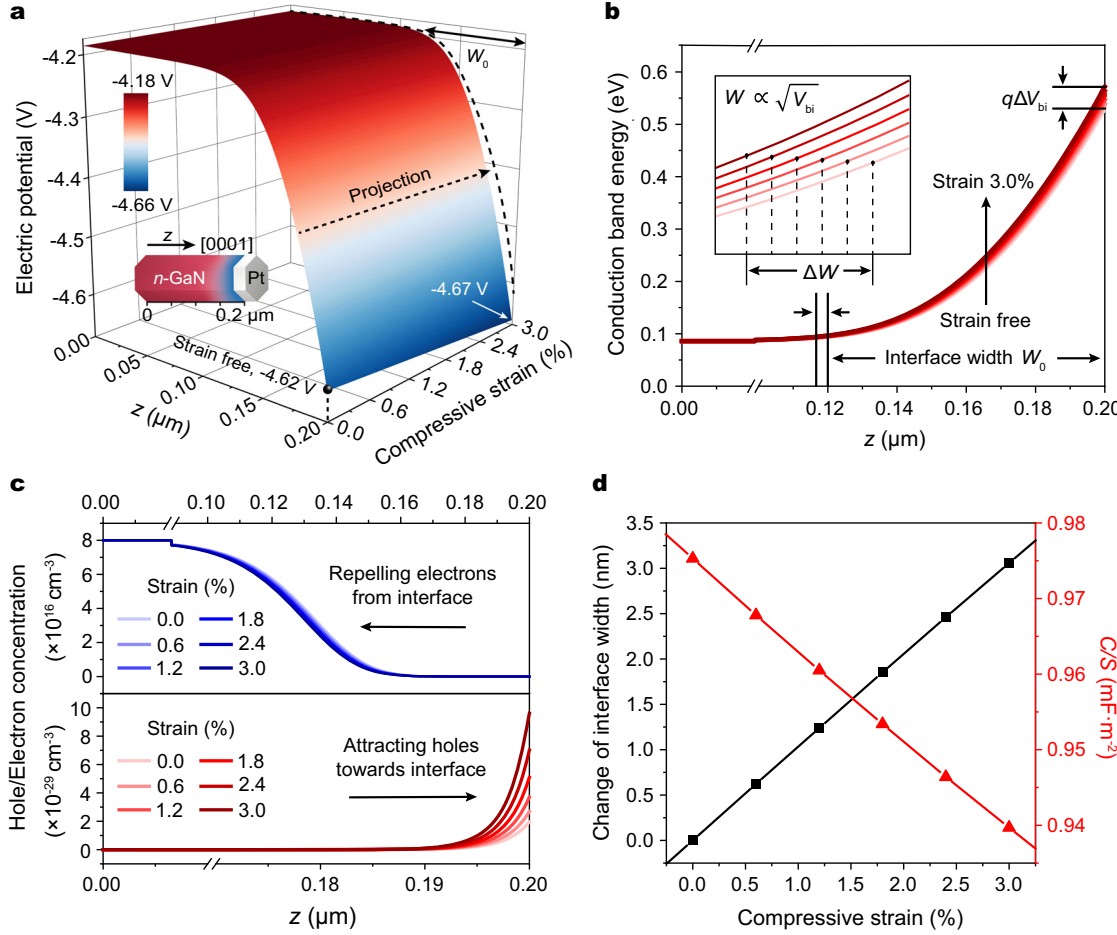

**Fig. 2 | Theoretical simulations of capacitive piezotronic effect. a** Calculated electric potential distribution of the Pt/$n$-GaN Schottky junction under compressive strain along the [0001] orientation of GaN. The dashed curve is the projection of the $V$–$z$ curve at strain free condition. **b** Calculated conduction band energy level and the corresponding change in interface width of the Schottky junction under compressive strain. $V_{bi}$ represents built-in potential. **c** Electron (upper part) and hole (lower part) concentration distribution at the interface of the Schottky junction under different compressive strain conditions. Electrons move away from the interface, and holes move towards the interface as the compressive strain increases. **d** The interface characteristics of the Schottky junction under compressive strain. The interface width presents a positive correlation with applied strain (black line), while the junction capacitance per unit area ($C/S$) exhibits a negative correlation with applied strain (red line).

charges will be produced at the interface, resulting in a reduction of the electric potential near the interface (Fig. 2a and Supplementary Fig. 1). This modification in electric potential induced by piezoelectric polarization will in turn increase the conduction band energy level near the interface, and hence raise the built-in potential ($V_{bi}$) of the Schottky junction. It is noted that the interface width is proportional to the square root of $V_{bi}$ at zero reverse bias ($V_R = 0$)[2],

$$W = \sqrt{\frac{2\varepsilon_s(V_{bi} + V_R)}{qN_d}} = \sqrt{\frac{2\varepsilon_s V_{bi}}{qN_d}}, \tag{3}$$

where $q$ is the elementary charge, and $N_d$ is carrier concentration. Accordingly, the interface width also exhibits an increment ($\Delta W$) with the increased compressive strain (Fig. 2b). Generally, the polarization field can increase the conduction band energy level, repelling the electrons away while attracting the holes towards the Schottky interface (Fig. 2c). As a result, the electron concentration decreases while the hole concentration increases near the interface. Since electrons are the majority carrier of $n$-GaN, the decreased electron concentration near the Schottky interface will induce more charged donor atoms in the depletion layer, resulting in an increased interface width. Specifically, with the compressive strain changed from 0% to 3%, the interface width is increased by ~3.1 nm and the corresponding junction

capacitance is decreased by ~0.04 mF/m² (Fig. 2d). These results show that redistribution of carriers by inner-crystal piezoelectric polarization can modulate the interface width and hence the junction capacitance, strongly demonstrating the possibility of capacitive piezotronics in Schottky junction. It should be worth noting that the piezotronic modifications on the interface width and the junction capacitance have often been ignored in the past DC electronics, but are crucial in AC settings.

**Interface width engineering in a single Schottky junction via piezoelectric polarization.** To verify the above theoretical simulations, we fabricated two-terminal devices consisting of a single Schottky junction and an Ohmic contact on a piezoelectric $n$-GaN single crystal with low carrier concentration (Supplementary Fig. 2). Figure 3a illustrates the electrical measurement system that uses a DC voltage to apply a bias to the single Schottky junction and simultaneously uses a 2 kHz AC signal ($V_{AC}$) to conduct the capacitance measurements under such bias condition. Then, we characterized the capacitance versus bias ($C$–$V$) curves and the corresponding ($1/C$)²–$V$ curves in Fig. 3b. As the reverse bias increases, the junction capacitance gradually decreases and finally tends to a constant value, indicating that the interface width increases until the Schottky junction is completely depleted. As the forward bias increases, the interface width decreases and thus the

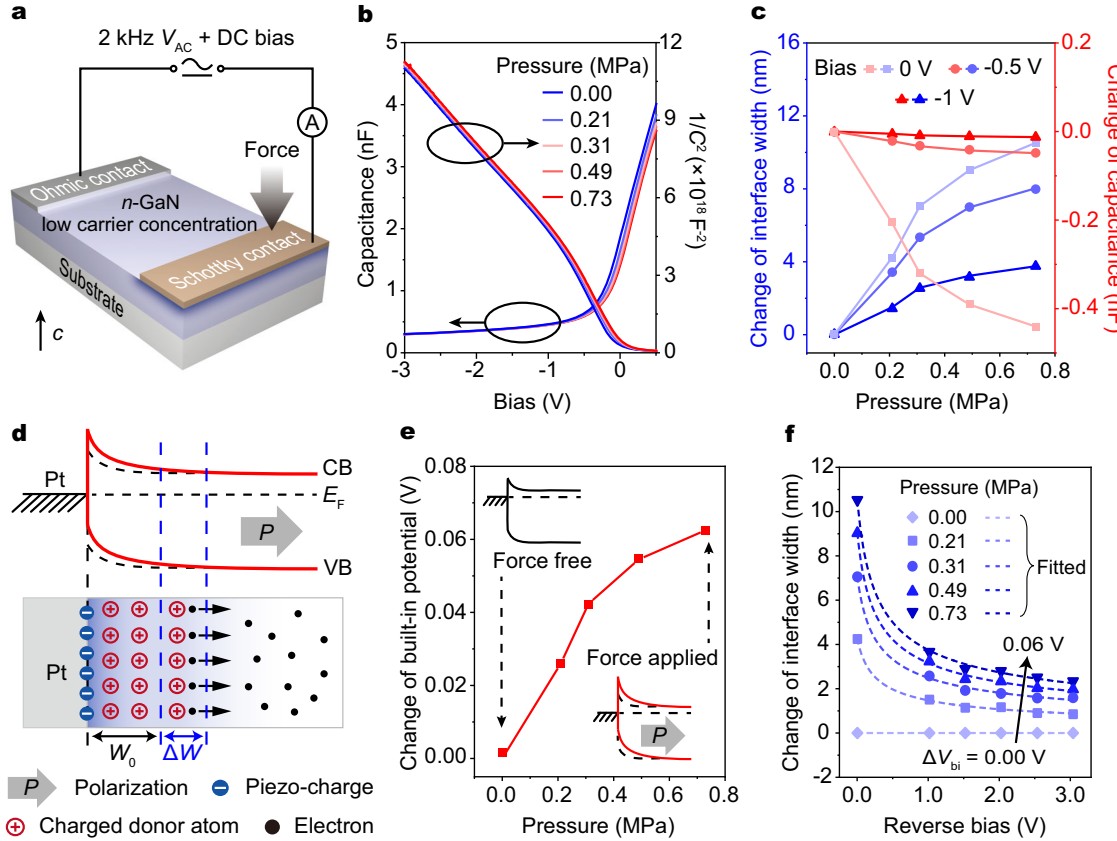

**Fig. 3 | Capacitive piezotronics in a single Schottky junction. a** Schematic of Pt/$n$-GaN Schottky junction with external loading forces along the $c$-axis of $n$-GaN. 2 kHz $V_{AC}$ and DC bias are simultaneously applied to the electrodes. **b** The $C$–$V$ characteristics and the corresponding $(1/C)^2$–$V$ characteristics of the device under different pressures. **c** Changes of junction capacitance and interface width as functions of applied pressures. Bias voltages are 0 V, −0.5 V, and −1 V. **d** Schematics show the detailed modulation process of the capacitive piezotronic effect in a single Schottky interface based on low-carrier-concentration GaN. The initial interface width $W_0$ is 81.0 nm at zero bias according to **b**. **e** The change of built-in potential as a function of applied pressures, which is calculated from the $(1/C)^2$–$V$ characteristics in **b**. **f** The change of interface width as a function of reverse bias under different pressures. The fitted curves (dashed lines) show that the influence of capacitive piezotronic effect on junction capacitance gradually degrades as reverse bias increases.

junction capacitance increases. As the applied bias gets close to the flat band voltage, the capacitance tends to infinity rapidly. The linear $(1/C)^2$–$V$ dependence at reverse bias also indicates the successful construction of the single Schottky junction, according to the formula of Schottky junction capacitance[2],

$$\left(\frac{1}{C_j/S}\right)^2 = \left(\frac{W}{\varepsilon_s}\right)^2 = \frac{2(V_{bi} + V_R)}{q\varepsilon_s N_d}. \tag{4}$$

External loading forces were then applied to the Schottky junction along the polarization $c$-axis of piezoelectric $n$-GaN, which were controlled by the displacement of a translation stage (Supplementary Fig. 3). As expected, when the loading force gradually increases, the interface width increases and simultaneously the junction capacitance decreases at the same bias (Fig. 3b, c). The change of junction capacitance reaches a maximum value of ~0.44 nF, and correspondingly, the interface width is increased by ~10.5 nm at a critical pressure of 0.73 MPa. The strain sensitivity ($\Delta C/p$) of the devices is measured in our experiments to quantitatively analyze the capacitive piezotronic modulation, which reaches the maximum of ~110 fF/mbar, and largely outperforms that of commercial capacitive pressure sensors ( ~ 0.1-0.7 fF/mbar[43,44], Supplementary Note 1). Figure 3d shows the detailed modulation process in an $n$-GaN-based single Schottky junction. Under loading force, the negative piezoelectric polarization charges

generated at the interface will repel electrons away from the interface and effectively enhance the built-in electric potential, which is demonstrated in Fig. 3e. According to formula (3), the enhancement in built-in electric potential further leads to an increase in the interface width, consequently reducing the junction capacitance. Besides, it should be noted that the bias has an influence on the capacitive piezotronic effect (Fig. 3c, f). To be specific, a high reverse bias makes the Schottky junction completely depleted, with its interface width and junction capacitance almost unchanged, thus degrading the influence of capacitive piezotronic effect on junction capacitance (Supplementary Note 3). Therefore, the devices exhibit stronger capacitive piezotronic modulation at low reverse bias, showing great potential in AC electronics with low power consumption and long lifetime.

## Capacitive piezotronics in dual Schottky junctions
To in-depth understand the capacitive piezotronic effect, we further investigated the force-dependent $C$–$V$ characteristics of dual Schottky junctions by fabricating a two-terminal device based on back-to-back Schottky junctions (Supplementary Fig. 2). The electrical measurement system in this experiment utilizes a DC voltage to apply bias between junction L and R, which is respectively reverse-biased (L) and forward-biased (R) at negative DC voltage, while overturned at positive DC voltage, and simultaneously utilizes a $V_{AC}$ signal to conduct capacitance measurements under such bias conditions. As shown in Fig. 4a, the force-induced piezoelectric polarization charges at one junction will increase its interface width, thereby decreasing its capacitance.

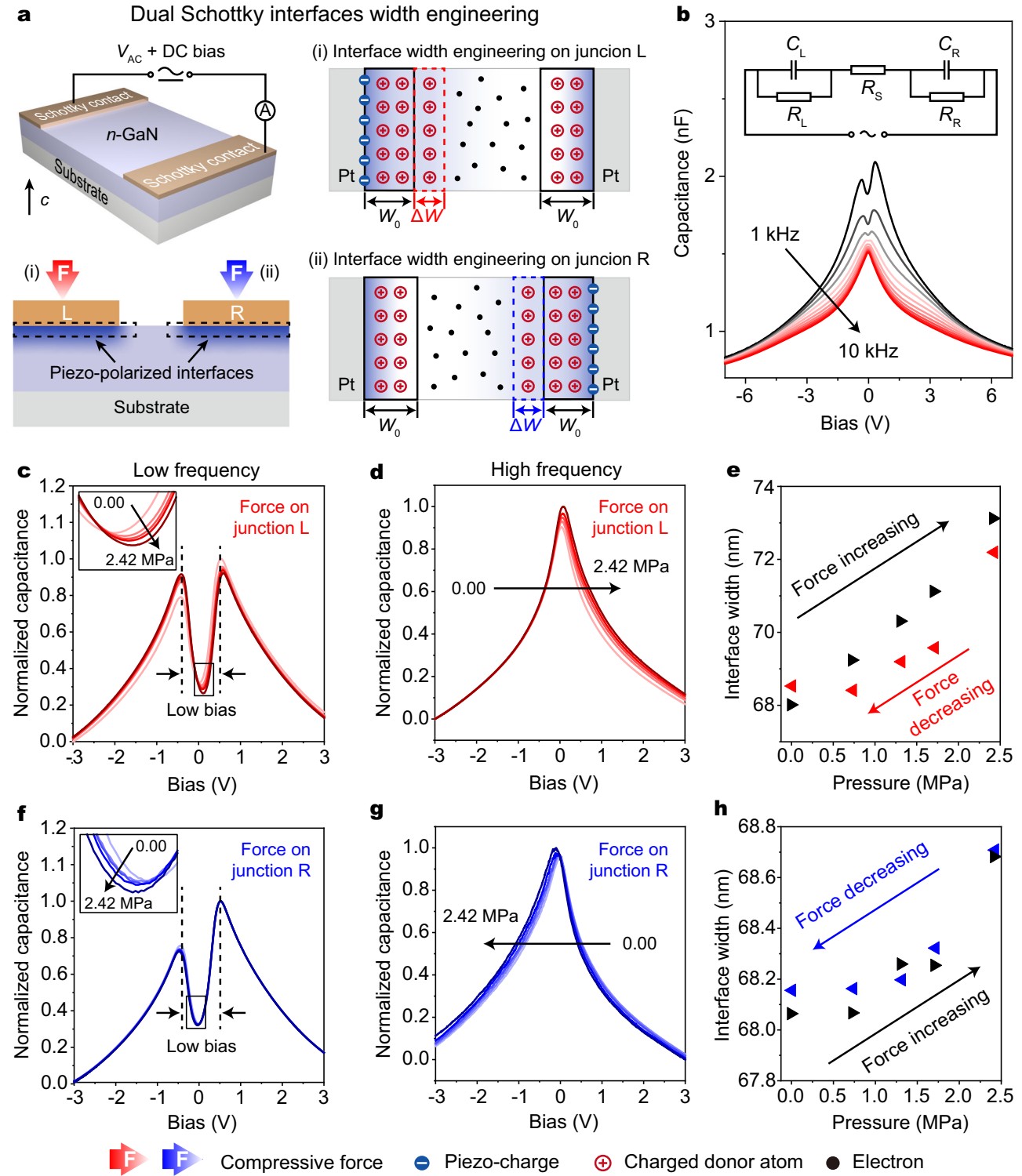

**Fig. 4 | Capacitive piezotronics in dual Schottky junctions. a** Mechanism of piezoelectric polarization to modify the interface width in dual Schottky junctions with force on junction L (i) and R (ii). **b** Dispersive $C-V$ characteristics of dual Schottky junctions. The inset presents the equivalent circuit of dual Schottky junctions, where $R_S$ represents the series resistance. **c**, **d** The $C-V$ characteristics of dual Schottky junctions based device under 1 kHz (**c**) and 10 kHz (**d**) AC settings with force on junction L. **e** The variation of interface width versus pressure corresponding to **c**. Respectively, **f**, **g** show the $C-V$ characteristics of a dual Schottky junction-based device under 1 kHz (**f**) and 10 kHz (**g**) AC settings, with forces on junction R. **h** The variation of interface width versus pressure corresponding to **f**. $W_0$ is 68.1 nm at zero bias. Results were acquired under identical test conditions and normalization conditions.

The piezotronic polarization is spatially confined to the stressed junction (Supplementary Fig. 4). This will lead to an asymmetric modulation on the dual Schottky junction width via the capacitive piezotronic effect.

Firstly, we measured the $C-V$ curves of dual Schottky junctions without loading force (Fig. 4b), and found two peaks (one around −0.5 V and the other around 0.5 V) at low-frequency ( < 5 kHz), while only one peak (around 0 V) at high-frequency ( ≥ 5 kHz). It can be seen

that the two peaks at low-frequency are slightly different, resulting from the fabrication-induced non-idealities of the dual Schottky junctions (e.g., local variations in effective doping concentration or slightly different metal pad areas). As the frequency increases, the junction resistances have a diminishing effect on the equivalent capacitance of dual Schottky junctions, so that the two peaks decrease until they degenerate into one peak (Supplementary Note 2). Under a force located at the junction L, one of the dual peaks at low-frequency gradually increases while the other decreases, and the $C-V$ curve at low bias region (approximately from −0.5 V to 0.5 V) exhibits a positive offset of 0.11 V with a decrease in the valley point (Fig. 4c). Moreover, an asymmetric phenomenon is also observed at high-frequency condition. The peak value of $C-V$ curve increases steadily, and the curve shifts toward positive bias (Fig. 4d). Next, we turned to apply the loading forces on the other Schottky junction, and observed completely overturned asymmetric modulation (Fig. 4f, g). These results further indicate that the asymmetric $C-V$ modulations indeed come from the strain-induced piezoelectric polarization. Furthermore, because the valley point occurs when the dual junctions are balanced, the valley capacitance could reflect the variation of each junction. Accordingly, we can obtain the changes of interface width of dual Schottky junctions from the change in the valley capacitance (Supplementary Note 4), which respectively reach 5.1 nm (Fig. 4e) and 0.6 nm (Fig. 4h) at 2.42 MPa. The smaller variation in the interface width with force on junction R could be attributed to the higher initial doping concentration near junction R. In order to quantitatively analyze the performance of the capacitive piezotronic device, we calculated the strain sensitivity ($\Delta C/p$) and found it as high as 9.0 fF/mbar at −0.5 V, which is much larger than that ( ~ 0.1-0.7 fF/mbar[43,44]) of commercial MEMS capacitive pressure sensors (Supplementary Note 1).

The force-dependent variation in $C-V$ curves arises from the asymmetric modulation on dual Schottky junctions via the capacitive piezotronic effect. In the absence of an external loading force, both the low-frequency and high-frequency $C-V$ curves of the dual Schottky junctions are centered at a near-zero bias region. When a loading force is applied to only one of the Schottky junctions, the strain-induced negative piezoelectric polarization charges increase both the built-in electric potential and the interface width of the stressed junction. This asymmetric stimulation requires an additional bias to compensate for the difference in built-in electric potentials, resulting in an obvious shift of the $C-V$ curves. Under low frequencies, taking junction L under loading force as an example, the entire $C-V$ curve shifts toward the positive direction to compensate for the asymmetric potential (Fig. 4c). Consequently, the peak in the negative bias region shifts closer to zero bias. Note that the peak capacitance of $C-V$ curve in low frequencies is primarily dominated by the interface width of the reverse-biased junction (junction L in the negative voltage region and junction R in the positive region). Thus, the peak shifting closer to zero bias is dominated by junction L. An anomalous phenomenon is observed that this peak capacitance increases rather than decreases as the junction L is stressed, which can be attributed to two mechanisms. On the one hand, although the negative polarization charges increase the interface width of the stressed junction, the peak moves to a lower bias owing to the shift of $C-V$ curve, and the lower bias will simultaneously decrease the interface width and hence compensate for the capacitance reduction. On the other hand, the changed effective carrier concentration near the interface enables interfacial electrostatic equilibrium (Fermi Level alignment) to be sustained within a narrower interface width, which also makes a contribution to the increasing capacitance (Supplementary Note 4). For the other peak of $C-V$ curve in the positive bias, the interface width of the unstressed junction R becomes the dominant factor. Since the piezotronic polarization is spatially confined to the stressed junction L, the interface width of the junction R is only influenced by the external bias. As this peak shifts to a higher bias region, the increased bias expands its interface width and thus decreases the peak capacitance. Conversely,

when the force is applied to junction R, the peak value in the negative bias decreases while the other peak value increases (Fig. 4f). This is a unique asymmetric capacitance modulation induced by the capacitive piezotronic effect (Supplementary Figs. 5 and 6). Under high-frequency settings, the capacitance of the peak at near-zero bias is dominated by both of the dual Schottky junctions. The interface widths of dual Schottky junctions exhibit opposing variations with one of the junctions stressed: the stressed junction widens, whereas the unstressed junction narrows. The experimental results further suggest that the narrowed junction may play a dominant role (Fig. 4d, g and Supplementary Fig. 7), leading to an increase in peak capacitance under loading forces. Furthermore, this increase is also ascribed to the variation in the effective carrier concentration near the stressed junction (Supplementary Note 5). These results show the strong modulation of the capacitive piezotronic effect on dual Schottky junctions, demonstrating significant potential for mechanically controlled high-frequency AC electronics.

## Universal nature of capacitive piezotronics in other piezoelectric semiconductors

In practice, the presence of free carriers in a piezoelectric semiconductor will generate an electrostatic screening effect on the produced piezoelectric polarization[28]. Nevertheless, we still observed a remarkable capacitive piezotronic effect in low-carrier-concentration GaN-based devices due to the partial screening of piezoelectric polarization. However, the screening effect of free carriers will be substantial in heavy-doped piezoelectric semiconductor. The proposed mechanism of capacitive piezotronics shown above may therefore be invalid. We conducted the same experiments on dual Schottky junctions using heavy-doped GaN (Supplementary Note 6 and Supplementary Figs. 24, 25). Completely different from the phenomenon of low-carrier-concentration GaN, excess capacitance is generated in $C-V$ characteristics due to the presence of interfacial traps[46]. No matter which junction the loading force is applied to, the excess capacitance is enhanced gradually without asymmetrical characteristics. This phenomenon mainly arises from the strain-induced bandgap tuning of the semiconductor that reduces the Schottky barrier height of the stressed junction.

To show that the capacitive piezotronic effect is widely present in piezoelectric semiconductors, a series of similar experiments was performed in a single crystal of ZnO, which is an ideal material with high piezoelectricity and semiconducting properties. Supplementary Fig. 8 shows the $C-V$ characteristics of the ZnO micro-wire based devices under tensile and compressive strain respectively, exhibiting a same trend as those of GaN with low carrier concentration (Fig. 4c). The $C-V$ curve presents a consistent shift towards negative bias and an increasing extremum under tensile strain, while shows a consistent shift towards positive bias and a decreasing extremum with the increased compressive strain. Moreover, the magnitude of tuning in ZnO micro-wire-based devices by the capacitive piezotronic effect seems to be more effective than that of bulk GaN. This is possibly owing to the fact that more effective piezoelectric polarization is generated within ZnO micro-wire resulted from a much larger strain as well as a higher piezoelectric constant. These results further demonstrate the effectiveness and universality of the capacitive piezotronic effect in piezoelectric semiconductors, which is promising for future mechanically tunable AC electronics and wide range applications.

## Capacitive piezotronics for communication systems

The junction capacitance of semiconductor devices (e.g., MS junction, MIS junction, $p-n$ junction) plays a crucial role in high-frequency transport and communications. Utilizing the capacitive piezotronic effect, external mechanical stimuli can precisely tune the junction capacitance, and hence enable frequency modulation and filtering of electrical signal during antenna transductions (Fig. 5a). The equivalent circuit of frequency modulation is schematically shown in Fig. 5b, comprising a resonant circuit made of an inductor ($L$) and a capacitive

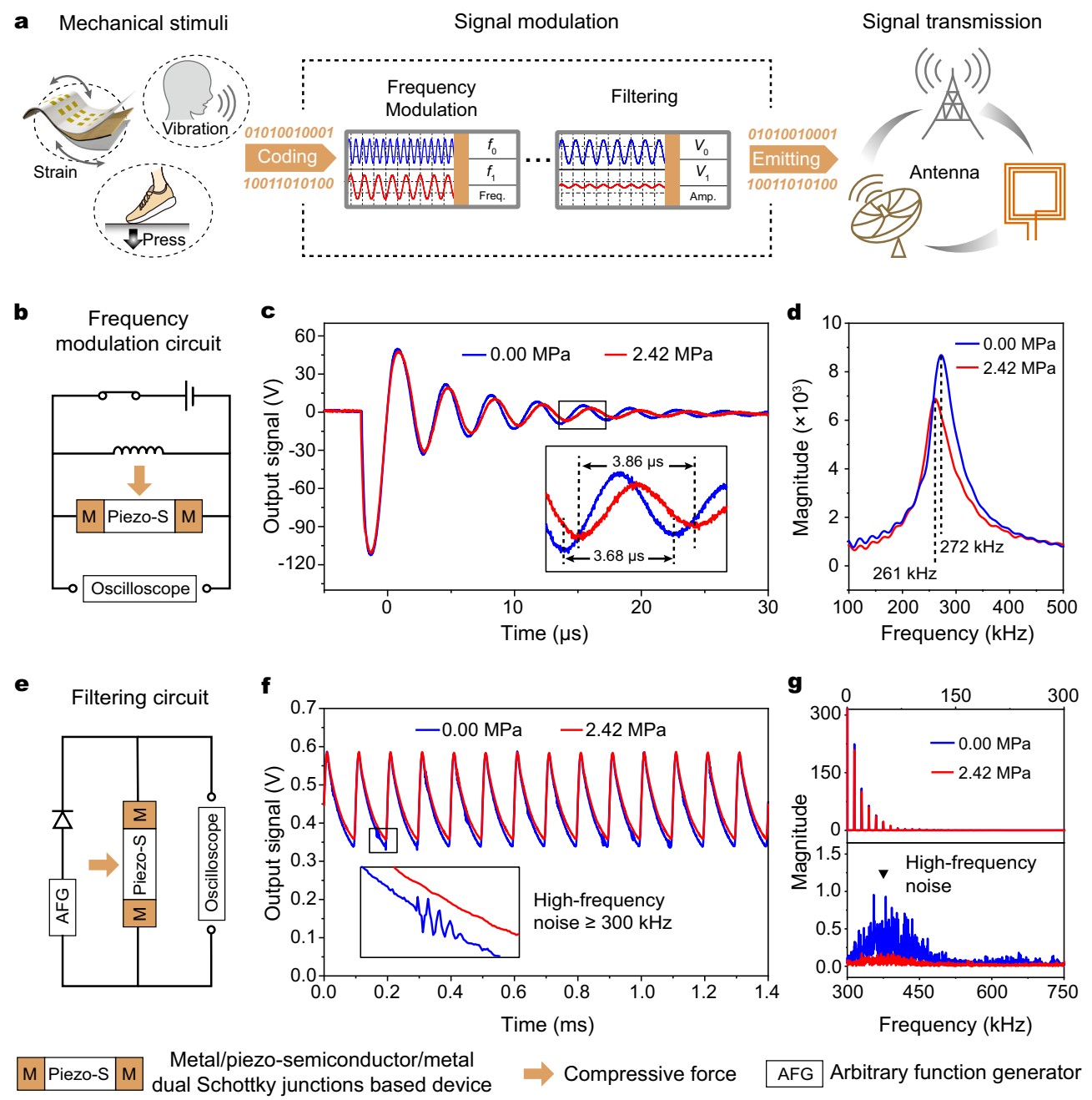

**Fig. 5 | Capacitive piezotronics for communication systems. a** Schematics show that the transmission of mechanical signals to the antenna involves mechanical stimuli sampling, signal coding, frequency modulation and filtering. **b** Frequency modulation circuit based on capacitive piezotronic effect. The inductance in the resonant circuit is 680 μH. **c** The oscillation signal of the circuit illustrated in (**b**) under loading force and force-free conditions. **d** Fast Fourier Transform spectrum of the oscillation signal in **c** shows a substantial change in oscillation frequency of the resonant circuit under loading force. **e** A low-pass filtering circuit (LPFC) based on capacitive piezotronic effect. The frequency of the initial sinusoidal wave signal is 10 kHz. **f** Filtered signal of the LPFC illustrated in **e** under loading force and force-free conditions. The high-frequency noise was filtered out under the loading force. **g** Fast Fourier Transform spectrum of the filtered signal shown in (**f**). The high-frequency noise (> 300 kHz) can be completely filtered out under loading force.

piezotronic device with dual Schottky junctions (serving as a tunable capacitor $C'$). The resonant frequency $f$ of this circuit is governed by:

$$f = \frac{1}{2\pi\sqrt{LC'}}. \tag{5}$$

Repetitive measurements on oscillation signals of the circuit were performed under force free and a loading force of 2.42 MPa on the capacitive piezotronic device (Supplementary Fig. 9). Upon loading

force, the junction capacitance of the capacitive piezotronic device is enlarged, and hence extends the oscillation period from 3.68 μs to 3.86 μs, corresponding to an oscillation frequency shift of 11 kHz (Fig. 5c, d). This strain-induced frequency modulation verifies a possible approach based on the capacitive piezotronic effect for signal processing in communication systems.

To further investigate the capacitive piezotronic effect for electrical signal filtering, we designed a low-pass filtering circuit (LPFC): a 10 kHz sinusoidal wave signal was first rectified by a commercial Schottky diode, and then was filtered by the capacitive piezotronic

device (serving as a tunable capacitor) (Fig. 5e). The filtered signals of LPFC were systematically carried out under force free condition and a loading force of 2.42 MPa (data reproducibility are presented in Supplementary Fig. 10). Our results show that the sinusoidal wave signal can be partially filtered by the capacitive piezotronic device without external loading force, but there is still some high-frequency noise. When the loading force was applied, the change in junction capacitance of the capacitive piezotronic device resulted in a reduction in its cutoff frequency, and a progressive degradation of high-frequency noise (Fig. 5f). These observations were also verified by the Fast Fourier Transform (FFT) spectrum: the low-frequency signal ($\leq 300$ kHz) remains preserved, while the high-frequency noise ($> 300$ kHz) can be completely filtered out by the device under loading force (Fig. 5g and Supplementary Figs. 11, 12). This phenomenon demonstrates the excellent performance of capacitive piezotronics on electrical signal filtering, providing a possibility for unconventional mechanical switching AC devices through dynamic tuning of junction capacitance.

## Discussion

In summary, we have demonstrated a strong capacitive piezotronic effect in piezoelectric semiconductors, in which the interface width and hence the junction capacitance at high-frequency AC settings ($\sim$ kHz-MHz) can be dynamically modulated by strain-induced piezoelectric polarization. Highly sensitive capacitive piezotronics for junction capacitance tuning have been achieved, of which the sensitivity largely outperforms that of commercial capacitive pressure sensors. This effect reveals a seldom-recognized coupling between piezoelectric polarization and high-frequency AC electronics, which enables mechanical tuning of communication systems, including signal transmission and filtering.

Based on its principle, the capacitive piezotronics is widespread in noncentrosymmetric semiconductors, including not only wurtzite structured ZnO and GaN, but also two-dimensional van der Waals materials ($MoS_2$, $In_2Se_3$, etc.) and perovskite materials. More likely, it can be extended to centrosymmetric semiconductors by strain-gradient induced flexoelectric polarization. Thus, our findings contribute to the development of mechanically tunable AC electronics in semiconductors, which are promising for unconventional AC electronic devices and electromechanical applications. This study also has the potential to stimulate further research on polarization-controlled AC technology, such as enhanced energy exchange and communications, AC piezoelectric/flexoelectric-phototronics, etc.

## Methods

### Materials

The [0001]-oriented single crystal GaN epitaxial wafers were purchased from Suzhou Nanowin Science and Technology Co., Ltd., which were grown on $c$-plane single crystal sapphire substrates via metalorganic chemical vapor deposition (MOCVD). The specific parameters of GaN epitaxial wafers (provided by Suzhou Nanowin Science and Technology Co., Ltd.) were as follows, dimensions: diameter: 50.8 mm ± 0.2 mm; thickness ≈4.5 μm; conduction type: $n$-type; resistivity (300 K) < 0.5 Ω·cm (carrier concentration $\leq 2 \times 10^{17}$ cm$^{-3}$) and <0.05 Ω·cm (carrier concentration $>1 \times 10^{18}$ cm$^{-3}$);

### Fabrication of capacitive piezotronic devices

The GaN epitaxial wafers were all cut into 1.0 cm × 1.0 cm squares. The sample pretreatment procedure consisted of sequential ultrasonic degreasing in acetone, absolute ethanol and distilled water (each for 15 min), followed by surface oxide layer removal using diluted hydrochloric acid (HCl) and drying under a nitrogen atmosphere. High-vacuum E-beam, magnetron sputtering and thermal evaporation (Kurt J. Lesker PRO Line PVD 75 and Denton Explorer 14) were used for electrode fabrication. Ti/Al/Ni/Au composite layers were deposited for Ohmic contacts, and Pt was deposited to construct a Schottky barrier.

Circular electrodes with a diameter of 1 mm were formed using lithography mask technology. The connection between copper lead wires and electrodes employed high-conductivity silver paste, and the thermal curing was completed in a vacuum environment (60 °C, −100 kPa).

### Capacitive piezotronic measurements

The capacitive piezotronic modification on the devices was performed using a semiconductor device parameter analyzer (Keysight B1500A) equipped with a Multi Frequency Capacitance Measurement Unit (MFCMU) module. In the MFCMU module, a small $V_{AC}$ with an amplitude of 200 mV and a frequency ranging from 1 to 10 kHz was applied to the capacitive piezotronic devices. Simultaneously, a sweeping DC voltage with a step size of 6 mV was used to apply a bias to the devices. The equivalent parallel circuit model ($C_p$) was adopted to obtain the capacitance of these devices. External forces were applied via a high-resolution (10 μm) translation stage, with the force-displacement relationship quantitatively monitored in situ using a calibrated pressure sensor.

### Signal modulation measurements

For frequency modulation measurements, an arbitrary function generator (Tektronix AFG31000) was used to provide charges for the resonant circuit, which includes a capacitive piezotronic device and a 680 μH inductor. The resonance signal of this circuit was measured by a mixed-domain oscilloscope (Tektronix MDO3024). For the filtering circuit, a 10 kHz sinusoidal wave signal was also provided by the arbitrary function generator and then half-wave rectified by a commercial Schottky diode. The half-wave rectified signal was filtered by a capacitive piezotronic device, and the filtered signal was also measured by the mixed domain oscilloscope.

### Theoretical calculations

The calculation on force-induced deformation in devices was performed with the solid module of the COMSOL program. The Young's modulus of GaN film and Sapphire is 324 GPa and 335 GPa, respectively. The Poisson's ratio of GaN film and Sapphire is 0.20 and 0.25, respectively. The calculation on capacitive piezotronics was performed with the AC/DC and Semiconductor module of the COMSOL program package through a 3D model of 200 nm wurtzite Ga-polar (0001) $n$-GaN. The top endpoint was set as a Schottky contact, while the bottom endpoint was set as an Ohmic contact and grounded. The intrinsic piezoelectric $c$-axis points to the Schottky interface, and the force is applied along the $c$-axis. The constants of GaN for numerical simulation were the relative dielectric constant $\varepsilon_r = 8.9$, the mobility of electrons $\mu_n = 1000$ cm$^2$V$^{-1}$s$^{-1}$, and the mobility of holes $\mu_p = 200$ cm$^2$V$^{-1}$s$^{-1}$. The electron affinity $\chi$ adopted here was 4.1 eV and the band gap $E_g$ was 3.39 eV. All the material coefficients above are available in the COMSOL Multiphysics material library. The temperature was 293.15 K, and the carrier concentration was about $8 \times 10^{16}$ cm$^{-3}$. The piezotronic effect was considered with the assumption that the piezoelectric charge was distributed uniformly within a very thin layer of $W_{piezo}$. With these conditions applied, sweeps of strain were conducted at zero bias using a stationary solver with a normal physical-control mesh to demonstrate the potential distribution, conduction band energy and carrier concentration distribution. With the above conditions, the change of interface width and capacitance with increasing strain could be calculated.

## Data availability

Source data are provided with this paper. The data that support the findings of this study are provided in the Source Data file.

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

## Acknowledgements

This research was supported by the Beijing Municipal Natural Science Foundation (Z230024[Hu]), the National Natural Science Foundation of China (Grant U25A6011[Wang], 52202162[Wang], 52192610[Wang], 52472164[Liu]), and the Program of Beijing Huairou Laboratory (JQ2023002A[Wang]).

## Author contributions

L.F.W., Z.L.W., and L.Y.X. conceived the project. L.F.W., L.Y.X., and Z.Z.Z. designed the experiments. L.Y.X., Z.Z.Z., and G.B.W. performed most of the experiments, and Y.X.H. contributed to the experiments. L.Y.X., Z.Z.Z., G.B.W., J.Y.Z., W.G.H., S.H.L., and L.B.C. analyzed the results.

L.Y.X. wrote the initial manuscript. L.F.W., Z.L.W., and S.H.L. revised the manuscript. All authors have discussed the results, commented on the manuscript, and approved the final version of the manuscript.

## Competing interests

The authors declare no competing interests.
