## [Transparent Peer Review file · Nature Communications]

Capacitive piezotronics

Corresponding Author: Professor Zhong Lin Wang

Version 0:

Reviewer comments:

Reviewer #1

(Remarks to the Author)

The article is devoted to the study of a new phenomenon called the capacitive piezotronic effect that occurs at semiconductor-piezoelectric / metal interface under pressure and voltage application. Although the magnitudes of these effects are relatively small, their practical application holds promise for enhancing the performance of devices where pressure sensitivity is crucial, such as stress sensors.

However, there are several issues that make it difficult to understand the presented results and the text.

1. The terminology and physical principles underlying this work are the first key questions.

1.1. The authors use the term "interface width." This term is used in physics and electronics, but can mean anything: it can ambiguously refer to structural, chemical, or electrical characteristics at the junction. In electronics, it typically refers to the roughness or length of the transition between two different materials in a heterostructure, or the atomic or chemical abruptness at the heterojunction or heterointerface (i.e., how many monolayers the transition from one material to another occurs over).

The authors do not provide a clear definition of the term in question. Furthermore, the interface width for the unperturbed transition is only indicated in Figure 8 of the Supplementary Information, where it is 68 nm at zero pressure. The Supplementary Information further states:

Line 65 "..... The piezotronic effect was considered with the assumption that the piezoelectric charge was distributed uniformly within a width of W_{piezo} ."

What is W_{piezo} ? This term is mentioned only once in the manuscript. In all relevant figures (e.g., Figs. 3d and 4a in the main text), it is represented as a single atomic layer or at least significantly narrower than the overall interface width. If W_{piezo} indeed corresponds to a single atomic layer, the question arises as to how piezoelectric charge can be uniformly distributed over such a thin region. Clear and precise definitions of terms such as W_{piezo} and interface width are crucial. Incorrect use or ambiguity of these definitions can fundamentally alter the underlying models and, consequently, the results presented in the paper.

1.2. The term "capacitive piezotronics" is quite beautiful. However, although it appears in the title and the first sentences of the abstract, the authors only provide a full explanation starting on line 109. Consequently, readers of the preceding 108 lines may not understand the precise meaning and scope of the concept being discussed.

Furthermore, this term is not entirely new. Commercial devices based on "capacitive piezotronics," such as the PCB3801 accelerometers, appeared on the market as early as 2005. Although these early devices were based on different materials, it is worth mentioning this historical context to clarify the relationship of this work to earlier ones.

1.3 From the very beginning of the manuscript, it should be clearly stated that the study is devoted to piezoelectric semiconductors. In the current version, this fundamental definition appears only in line 122.

1.4. Care must be taken to ensure that all variables described in the descriptive sections of the text (voltage, for instance) are consistently and accurately related to the precise physical quantities and notations used in the formulas and figures.

2. The second key questions are devoted to presenting description of experimental results and experiment.

2.1 I believe the manuscript lacks a clear physical representation of the device under study. In addition to schematic diagrams, it would be helpful to include a more specific illustration or description demonstrating realistic pressure application to the device. Furthermore, I would appreciate a photograph or visual representation clearly demonstrating:

Supplementary, Line 49: "External forces were applied via a high-resolution translation stage, with the force-displacement relationship quantitatively monitored in situ using a calibrated pressure sensor"

2.2. The supplementary materials contain numerous virtually identical experimental curves (capacitance versus voltage), making them difficult to understand. Furthermore, frequent references to 20 figures in the main text significantly complicate reading and disrupt the flow. I propose reorganizing the article so that the main text remains self-contained and easily

understandable. Instead of citing supplementary materials multiple times on each page of the main article, it would be more effective to structure the supplementary materials so as to clearly indicate which sections or figures provide additional explanations or confirm specific parts of the main text.

2.3 What is the statistics of the results? How many samples were studied?

2.4. The manuscript contains several examples of the use of non-technical or indefinite adjectives:

Supplementary:

Line 48 a precisely controlled DC bias ...

Line 50 ... high-resolution translation stage...

Line 145 we can get the low-frequency (ω -low) and high-frequency 146 (ω -high) capacitance...

Line 242 ...as the frequency increases, the excess capacitance response should be weakened because the carrier capture/release rate is too slow to keep up with the high-frequency signals...

Main text

Line 95 ...to precisely control the junction capacitance under high frequency AC settings

Line 174 ... can be precisely controlled by the displacement ...

.....

And more....

It is important that these qualitative terms be accompanied by specific numerical values.

2.5. Error bars should be discussed for experimental results.

3. The use of the specific words.

3.1. Quite often the authors use the words, that in principle may mean what maybe they mean, but in context of the paper they are confusing:

Line 114 "This dynamic hysteresis phenomenon is fundamentally equivalent to the capacitive reactance (XC) inherent in MS junction, resembling a charge-discharge process in a conventional capacitor."

Phenomenon (if consider dynamic hysteresis as a processes can be equivalent to the another process – charging-discharging). But it cannot not be equivalent to the value, such as capacitive reactance (XC).

Line 150 "... potential (Vbi) of the Schottky junction. It is noted that..." Who noted that?

Line 177 "...The linear correlation of (1/C)²-V curve ...". must be changed by : "The linear (1/C)²-V dependence..."

Line 195 "... built-in electric potential, which is intuitively demonstrated by the experimental results...". What means "intuitively" in the context of experimental demonstration?

Line 335 "...For the capacitive piezotronics, reactance (X) is applied to further describe the effect ...". How the measurable value (X) can be applied... to describe?

3.2. Quite often the adverbs are used instead of adjectives:

Line 85: "... Particularly, with widely use of... materials..."

And more...

4. It is also necessary to carefully review and correct the grammar throughout the manuscript.

In conclusion,

I need additional evidence which support the result (see #1 and #2 in my report).

It requires better data presentation and more logic and accuracy in the text.

The work demonstrates significant results and deserves publication after significant revision.

Reviewer #2

(Remarks to the Author)

General Assessment

The authors present a comprehensive investigation into how strain-induced polarization in piezoelectric semiconductors can be exploited as an interface engineering mechanism to modulate the interface width and the junction capacitance under AC operation in Schottky-diodes structures. The study combines finite element simulations and experimental measurements on single and dual GaN-based Schottky diodes to explore how controlled compressive strain affects the built-in potential, interface width, and capacitance, reporting strain sensitivities exceeding those of commercial capacitive pressure sensors. Furthermore, the authors illustrate the potential of these mechanisms in practical applications by demonstrating implementations in communication circuits such as frequency modulators and filtering systems.

The manuscript provides a substantial amount of material supporting the main claims, including physical descriptions, finite element simulations, and a broad set of experimental results. The mechanisms are interpreted with an appropriate level of detail, and the inclusion of practical applications and comparative analyses strengthens the overall study. Nevertheless, several aspects of the work would benefit from clarification and refinement to further improve its rigor and readability before it can be considered for publication.

General Revisions

It should be noted that capacitance variation in piezoelectric and piezotronic devices under mechanical stress has been reported previously; for instance, in bulk ceramics (Cho et al., JJAP, 1995 10.1143/JJAP.34.1591) and in ZnO-based Schottky structures (Zhang et al., Sensors 2021, 21(6), 2253; <https://doi.org/10.3390/s21062253>). In this respect, the present manuscript goes beyond those studies and offers a conceptual and practical advance.

Nevertheless, in light of the existing literature, the authors are encouraged to cite and briefly discuss these and others prior works, either in the Introduction or main text, and to compare their findings and underlying mechanisms where appropriate. In addition, they are encouraged to balance some novelty statements to ensure that the contribution is presented in a balanced and accurate way. Such refinements would further enhance the credibility of the manuscript and draw clearer attention to its genuine advances.

Abstract Revision

The abstract does not clearly specify the type of investigation conducted, whether the reported results are theoretical/simulative or experimental, nor does it explain how the numerical values were obtained. The authors should explicitly state that the study combines finite element simulations and experimental measurements, and that the reported numerical results refer to single and dual GaN-based Schottky devices. This clarification would provide a more accurate and transparent representation of the work.

Manuscript Revisions

1. In the caption of Figure 1a, please specify that “P” denotes the polarization vector.
2. In Figure 1b (ii-1, ii-2, ii-3), both compressive and tensile strains are illustrated as acting along the same direction of the c-axis. Since tensile and compressive strain correspond to opposite deformation directions, leading to opposite polarization vectors, showing both with arrows in the same direction is misleading. The schematic should be revised to accurately depict strain directions relative to the c-axis and the corresponding polarization orientations.
3. The authors indicate the parameters used in the simulations, but should clarify which crystallographic polarity (Ga-face or N-face) is considered and under what bias conditions the simulations were performed.
4. In addition to the schematics, an optical image of both the single and dual Schottky diodes should be provided. The authors should also specify which capacitor area “S” is used in equation (3).
5. Since the junction capacitance is analyzed across various bias ranges, including an I–V curve of the Schottky diode under strain-free conditions would provide a more complete picture of the device behavior.
6. The C–V measurements of the single Schottky diode under compressive strain are reported without indicating the frequency of the AC signal. The authors should specify the frequency used, justify their choice, and clarify whether the capacitance variation depends on frequency, as was done for the dual-Schottky configuration. This might be useful to better understand the dual-Schottky junction case.
7. Supplementary Figure 3 should be corrected, as it currently shows the same strain direction for both compressive and tensile loading.
8. In the section “Capacitive piezotronics in dual Schottky junctions”, the authors should specify on which junction the bias is applied and explain why they expect that loading one junction does not affect the interface width of the other (line 210-212). They should add a reference to this sentence.
9. To improve readability, the color scheme of Figure 4b should differ from the subsequent figures. In reference to the sentence at line 217 the authors could also specify the parameters (junction area,..) that make the two Schottky junctions different and explain how these differences lead to asymmetric C–V peaks. The two Schottky barriers, indeed, are considered symmetrical over the manuscript. Additionally, there appears to be an inconsistency: in Figure 4b the left peak is higher than the right one, while the opposite occurs in subsequent figures.
10. Figures 4c and 4f are not self-explanatory. It is unclear which curve corresponds to the strain-free case. The referee suggests replacing them with Supplementary Figures 8a and 8c, which are clearer. The authors should also indicate, at least in the caption, the low and high frequencies used for the strain-dependent measurements. Moreover, the rightward shift of the C–V curve under strain is evident near the valley point; the arrows should be repositioned accordingly.
11. Figure 4c shows that at low bias the C–V curve valley shifts to the right when the left junction is under strain. What do they associate with the leftward shift observed at higher bias?
12. Just for clarity, the authors could specify why a smaller capacitance variation is observed when the right junction is loaded at low frequency, in Figure 4f.
13. Supplementary Figure 9 does not seem consistent with the description provided in the manuscript. Why is the barrier height of the left junction in reverse mode and under strain (i) lower than in the strain-free condition? Why is the depletion width in that case smaller than in the strain-free configuration? When the strain is applied to the right junction, why is the left-junction barrier height higher than in the other two cases?
One would expect the strained junction in reverse mode to exhibit a higher barrier and wider depletion region. If these plots represent a different potential distribution rather than local barrier heights, the authors should clarify this explicitly. In that case, the statement around lines 210–212 regarding the independence of each junction under strain may need reconsideration.
14. The explanation of the asymmetric strain-induced modulation of the C–V peaks needs clarification. One would expect the strained junction in reverse bias to show lower capacitance than the strain-free case, since its depletion region widens. However, the observed peak shows higher capacitance. While the shift of the peak to lower negative biases the authors refer to in line 245-248 compensates for the widened barrier, it does not fully explain the higher capacitance compared to the strain-free situation. A more explicit discussion of the interaction between the two junction might be needed to explain why the capacitance increases despite the widened barrier.
14. In Supplementary Note 4, the role of R_s in the theoretical model should be clarified.
15. In Supplementary Note 5, the derivation of Equation (17) is not clear and a reference should be added.
16. The clarity of the manuscript could be further improved by adding additional details on the device fabrication, experimental procedures, and physical mechanisms directly within the main text, rather than leaving them mainly in the Supplementary Information.
17. The last section of the manuscript “ Relationship between capacitive piezotronics, resistive piezotronics and traditional capacitive effect “ could be summarized and added in the introduction or in the ‘ mechanism of the capacitive piezotronics’, since the main claims have already been introduced.

Version 1:

Reviewer comments:

Reviewer #1

(Remarks to the Author)

All comments have been taken into account in the new version, and the quality of the article has been significantly improved. I believe it can be published in its current form.

Reviewer #2

(Remarks to the Author)

The authors have responded thoroughly to the reviewer's comments. They have referred to prior literature, improved the clarity of several figures, added the requested derivations and theoretical models, and provided additional details in the main text. In particular, the physical interpretation of the C–V curves shape, the observed shift, and the asymmetric capacitance behavior is now clearer and better motivated. Overall, the manuscript has significantly improved. A few points, however, would still benefit from further clarification before publication.

Abstract revision

While the revised abstract more clearly states that both theoretical analysis and experimental investigations are involved, it remains unclear how the reported quantitative values are obtained and to which specific devices they refer. The authors are encouraged to explicitly state that finite-element simulations and experimental measurements were carried out on GaN-based (and subsequently ZnO-based) Schottky diodes, and to clarify which results are experimentally measured and which are supported by simulations. Phrases such as “Experimental measurements on GaN-based Schottky diodes show that...” and “Finite-element simulations confirm that...” could be considered.

Moreover, when referring to “high frequency” operation, the authors should provide indicative frequency ranges (e.g., kHz regime) in order to better contextualize the practical applicability of the devices.

Specific comments to the authors' responses

R1. Although the symbol “P” has been replaced with “polarization” in Fig. 1a, it is still used elsewhere without definition. For consistency, it should be explicitly defined or replaced throughout the manuscript.

R6. The authors have provided a convincing theoretical justification for the choice of measurement frequency and for the frequency-independent behavior of a single Schottky junction under reverse bias. While an explicit experimental verification of the absence of frequency dispersion would have been appreciated, the present explanation is sufficient and consistent with established literature.

R8. The clarification of the measurement setup and the demonstration that strain effects are spatially localized in the stressed junction are appreciated. However, in the low-frequency C–V characteristics of the dual Schottky junction (Figs. 4c and 4f), the reported zero-strain curves exhibit noticeably different peak asymmetries. The authors are encouraged to clarify whether these strain-free curves were acquired under identical measurement and normalization conditions, and whether factors such as mechanical boundary conditions, contact resistance, or low-frequency drift/hysteresis could account for these differences.

R9. The explanation of the asymmetric C–V peaks in terms of fabrication-related non-idealities (e.g., local variations in effective doping concentration or slightly different metal pad areas) is physically reasonable. The authors are encouraged to state this more explicitly in the main text, for instance by revising the sentence: “It can be seen that the two peaks at low frequency are slightly different, resulting from...” or by referring to this point in the Methods section.

R14. The revised description of the increase in the capacitance peak is clearer. The authors are encouraged to elaborate slightly more in the main text on the role of the effective charge concentration, while keeping the full quantitative details in the Supplementary Information. This would help convey a more complete physical picture within the manuscript.

Furthermore, although the valley in low-frequency measurements and the peak in high-frequency measurements are experimentally observed at zero bias (Figure 4), statements such as “the valley occurs only when $R_1=R_2$ ” (Supplementary Note 4.2) or “the peak locates at zero bias when the junctions are perfectly symmetric” (Supplementary Note 5) may be misleading in the presence of fabrication-induced asymmetries. It would be clearer to specify that the condition $R_1=R_2$ is satisfied at a specific bias point (which may coincide with zero bias), rather than implying perfect structural symmetry.

Finally, the authors are encouraged to further refine the language and avoid overly strong expressions such as “abnormal phenomenon”, “huge impact”, or “unexpectedly giant effect”, in order to maintain a balanced scientific tone.

Version 2:

Reviewer comments:

Reviewer #2

(Remarks to the Author)

The authors have adequately addressed all the comments in the revised manuscript. In my opinion, the paper can be accepted for publication in its current version.

Point-to-Point Response to Reviewers' comments:

We would like to thank the reviewers for their in-depth reviews and constructive suggestions regarding our manuscript. In the pages that follow, we have provided our responses to each of the reviewer's questions.

Reviewer #1 (Remarks to the Author):

The article is devoted to the study of a new phenomenon called the capacitive piezotronic effect that occurs at semiconductor-piezoelectric/metal interface under pressure and voltage application. Although the magnitudes of these effects are relatively small, their practical application holds promise for enhancing the performance of devices where pressure sensitivity is crucial, such as stress sensors.

However, there are several issues that makes difficult understanding of the presented results and the text.

Response:

We thank the reviewer for acknowledging the importance of our work. And we also thank the reviewer's detailed and responsible reviewing of our work.

Question #1

The terminology and physical principles underlying this work are the first key questions.

Response #1:

Thanks for the reviewer's constructive comment. A clear description of the terminology and physical principles is indeed crucial for understanding our work. We have made comprehensive revisions to the manuscript and supplementary materials in response to your subsequent Questions #1.1-1.4.

Question #1.1

The authors use the term "interface width." This term is used in physics and electronics, but can mean anything: it can ambiguously refer to structural, chemical, or electrical characteristics at the junction. In electronics, it typically refers to the roughness or length of the transition between two different materials in a heterostructure, or the atomic or chemical abruptness at the heterojunction or heterointerface (i.e., how many monolayers the transition from one material to another occurs over).

The authors do not provide a clear definition of the term in question. Furthermore, the interface width for the unperturbed transition is only indicated in Figure 8 of the Supplementary Information, where it is 68 nm at zero pressure. The Supplementary Information further states:

Line 65 “...The piezotronic effect was considered with the assumption that the piezoelectric charge was distributed uniformly within a width of W_{piezo} .”

What is W_{piezo} ? This term is mentioned only once in the manuscript. In all relevant figures (e.g., Figs. 3d and 4a in the main text), it is represented as a single atomic layer or at least significantly narrower than the overall interface width. If W_{piezo} indeed corresponds to a single atomic layer, the question arises as to how piezoelectric charge can be uniformly distributed over such a thin region. Clear and precise definitions of terms such as W_{piezo} and interface width are crucial. Incorrect use or ambiguity of these definitions can fundamentally alter the underlying models and, consequently, the results presented in the paper.

Response #1.1:

Thanks for the reviewer’s insightful comment. We are sorry that our descriptions are not clear. We have now provided precise descriptions of the terminology and revised them in manuscript and supplementary materials as follows.

(1) The authors use the term “interface width”. This term is used in physics and electronics, but can mean anything: it can ambiguously refer to structural, chemical, or electrical characteristics at the junction. In electronics, it typically refers to the roughness or length of the transition between two different materials in a heterostructure, or the atomic or chemical abruptness at the heterojunction or heterointerface (i.e., how many monolayers the transition from one material to another occurs over). The authors do not provide a clear definition of the term in question.

As is well known, the potential energy within the region near the interface (e.g., in p - n junctions, Schottky junctions or tunnel junctions) is higher than that of the surrounding space, forming an “energy barrier” region. Ideally, the interface width could be defined as the width of this region, while the interface barrier height is the energy required for electrons to cross the region.

Previous studies about piezotronics mainly focus on regulating the interface barrier height utilizing piezoelectric polarization, so as to control the carrier transport and thus change the “resistance” of DC devices. It can be concluded as “resistive piezotronics”. In fact, piezoelectric polarization could also simultaneously modulate interface width, which is closely associated with junction capacitance, the core of capacitive piezotronics proposed in our work.

In our experiments, the term “interface width (W)” specifically refers to the width of depletion layer (W_{de}) at Schottky interface, which is a well-established concept in semiconductor physics (Sze, S. M. *et al. Physics of Semiconductor Devices*. John Wiley & Sons, 2021), denoting the width of a region where the mobile charged carriers are completely depleted. This parameter directly affects the junction capacitance (C) through the relationship

$$\frac{C}{S} = \frac{\epsilon_S}{W_{de}}, \quad (\text{R} - 1)$$

where S is the area of interface and ϵ_S represents the dielectric constant of the semiconductor. Modulation on the depletion layer width and thus the junction capacitance plays a critical role in semiconductor device, especially for communication applications such as high-frequency transport and high-cutoff-frequency radio frequency diode.

The term “interface width” instead of “depletion layer width” is adopted in our work, as it provides a more proper and comprehensive description. Theoretically, the interface width includes not only depletion layer width but also the width of insulator/dielectric layer in MIS structures, the width of functional layer in some composite structures and so on. Also, the traditional theory about depletion layer may become invalid in low-dimensional materials such as nanowires and two-dimensional materials. Therefore, the term “interface width” is more universal in different device structures and materials, providing better insight into the mechanism of the capacitive piezotronics.

To provide a clear and precise description of “interface width”, we have revised the manuscript as follows:

- In line 78-80: *“In addition, the interface width of heterostructures, referring to the width of the region where the potential energy is higher than that in the surrounding space, also plays a critical role in semiconductor devices...”*
- In line 120-122 in section “Mechanism of the capacitive piezotronics”: *“In a typical Schottky junction, the interface width (W) specifically refers to the width of the depletion layer at Schottky interface, where the potential energy is higher than that in the surrounding space...”*

(2) Furthermore, the interface width for the unperturbed transition is only indicated in Figure 8 of the Supplementary Information, where it is 68 nm at zero pressure.

Thank the reviewer for this comment. The following revisions are aimed at clarifying the interface width without strain in manuscript:

- In the caption of Figure 3d (line 481-482): *“... W_0 represents the initial interface width without force (81 nm at zero bias) and ΔW represents the change of interface width under loading force...”*
- In the caption of Figure 4a (line 490-491): *“... W_0 represents the initial interface width without force (68 nm at zero bias) and ΔW represents the change of interface width under loading force...”*

(3) The Supplementary Information further states: Line 65 “...The piezotronic effect was considered

with the assumption that the piezoelectric charge was distributed uniformly within a width of W_{piezo} .” What is W_{piezo} ? This term is mentioned only once in the manuscript.

Generally speaking, W_{piezo} is the width of the region in which we hypothesis the piezoelectric charge uniformly distributes. The piezoelectric polarization induces opposite net charges respectively distributed at the two surfaces of the material. These net charges refer to piezoelectric polarization charges, which are non-mobile ionic charges adjacent to the interface. In simulations, the piezoelectric polarization charges are often assumed to be distributed uniformly within a very thin width, which is denoted as “ W_{piezo} ”. Since it is merely an assumption utilized in theoretical simulation, we have placed it only in section “theoretical calculations” of “Methods”.

(4) In all relevant figures (e.g., Figs. 3d and 4a in the main text), it is represented as a single atomic layer or at least significantly narrower than the overall interface width. If W_{piezo} indeed corresponds to a single atomic layer, the question arises as to how piezoelectric charge can be uniformly distributed over such a thin region. Clear and precise definitions of terms such as W_{piezo} and interface width are crucial. Incorrect use or ambiguity of these definitions can fundamentally alter the underlying models and, consequently, the results presented in the paper.

Firstly, we would like to clarify that the “ W_{piezo} ” doesn’t corresponds to only a single atomic layer. The piezoelectric polarization charges in all relevant figures are schematic presentations, intended only to clarify the direction of polarization at the interface. Similar presentations have also been applied in previous works on piezotronics (Liu, S. *et al. Adv. Mater.*, 2019, 31, 1905436; Yu Q. *et al. Nat. Commun.*, 2022, 13, 778).

In fact, “ W_{piezo} ” is indeed significantly narrower than the overall interface width. This is because that piezo-semiconductor also shows screening effect (though poor than conductor), thus the positive and negative charges tend to distribute within a small depth from the surface to get a lower electric potential energy. In nanodevices or even microdevices, the assumption that the polarization charge was distributed uniformly within a very thin W_{piezo} has also been proved effective in simulations and calculations (Zhang Y. *et al. Adv. Mater.*, 2011, 23, 3004).

Notably, although W_{piezo} is significantly narrower than the interface width, the polarization charges can still effectively modulate the interface characteristics. This is already confirmed by previous studies on “resistive piezotronics”, which have demonstrated that the strain-induced polarization charges have a huge impact on the migration and redistribution of carriers at the interface (Liu, S. *et al. Adv. Mater.*, 2019, 31, 1905436; Yu Q. *et al. Nat. Commun.*, 2022, 13, 778). To be specific, negative polarization charges can attract the holes to the interface while keep the electrons away, leading to the enhancement in interface barrier height and width. On the contrary, positive polarization charges induce completely overturned changes. In this way, the junction

resistance and capacitance can be modulated, which is the “resistive piezotronics” in previous works and “capacitive piezotronics” of this work, respectively.

We have provided a precise description of “interface width” and “ W_{piezo} ” and we hope that these revisions have resolved the ambiguity in terminology and strengthened the foundation of our physical model.

Question #1.2

The term “capacitive piezotronics” is quite beautiful. However, although it appears in the title and the first sentences of the abstract, the authors only provide a full explanation starting on line 109. Consequently, readers of the preceding 108 lines may not understand the precise meaning and scope of the concept being discussed.

Furthermore, this term is not entirely new. Commercial devices based on “capacitive piezotronics,” such as the PCB3801 accelerometers, appeared on the market as early as 2005. Although these early devices were based on different materials, it is worth mentioning this historical context to clarify the relationship of this work to earlier ones.

Response #1.2:

Thanks for the reviewer’s constructive and important suggestions.

(1) The term “capacitive piezotronics” is quite beautiful. However, although it appears in the title and the first sentences of the abstract, the authors only provide a full explanation starting on line 109. Consequently, readers of the preceding 108 lines may not understand the precise meaning and scope of the concept being discussed.

Thanks for the reviewer’s praise on the term “capacitive piezotronics” and we agree that its full explanation should be referred at a proper position in the manuscript. Although we highly respect the reviewer’s opinion, we would like to clarify that the explanation of “capacitive piezotronics” has been already presented before line 109 for several times:

- In the abstract of the original manuscript: “...*It utilizes the strain-induced piezoelectric polarization to precisely control the interface width of heterostructures and hence modulate the junction capacitance at a high frequency...*”
- In line 93-95 of the original manuscript: “*Here, we report the first study of capacitive piezotronics, showing that the inner crystal piezoelectric polarization can be used as a “gate” signal to reversibly tune the interface width of heterostructures, so as to precisely control the*

junction capacitance under high frequency AC settings...”.

We recognize that these two explanations are underemphasized in our original manuscript and we have revised them to give the emphasis they deserve:

- In line 41-45 in the abstract: “...*Here, we report the capacitive piezotronics, which utilizes the strain-induced piezoelectric polarization to precisely control the interface width of heterostructures and hence modulate the junction capacitance at a high frequency. It is a distinctive regulation mechanism that is widely present in piezoelectric semiconductors (ZnO, GaN, etc.)...*”. We highlight the core definition of the capacitive piezotronics to give more emphasis on the precise meaning and scope of the concept.
- In line 98-101: “*Here, we report the first study of capacitive piezotronics, demonstrating that polarization potential in piezoelectric semiconductor can be used as a “gate” signal to reversibly tune the interface width of heterostructures, so as to control the junction capacitance under high frequency AC settings...*”. We added the “piezoelectric semiconductor” in the sentence to further clarify the meaning and scope of the concept.

Moreover, we would like to provide a brief summary to help the reviewer understand our thought about introducing the concept of “capacitive piezotronics”. As the capacitive piezotronics is an expansion to the piezotronic effect, we think it would be more readable to introduce from the traditional resistive piezotronics and then propose the new effect by indicating the differences. Based on this aspect, we firstly introduce the context of polarization-driven interface engineering in semiconductors, highlighting the well-established resistive piezotronic effect that modulates interface barrier height and thus control the current/resistance of DC devices. After that, we demonstrate the critical role of interface barrier width in high-frequency AC electronics, and emphasizes the urgent need for achieving dynamic manipulation of the interface width. Accordingly, we change the focus of piezotronic effect from modulation on interface barrier height to that on interface barrier width. Based on these foundations, we could claim our core concept “capacitive piezotronics” reasonably.

(2) Furthermore, this term is not entirely new. Commercial devices based on “capacitive piezotronics,” such as the PCB3801 accelerometers, appeared on the market as early as 2005. Although these early devices were based on different materials, it is worth mentioning this historical context to clarify the relationship of this work to earlier ones.

We thank the reviewer for careful consideration on the historical context of our work. We would

like to clarify that the terms “capacitive piezotronics” and “capacitive piezotronic effect” have not been proposed before. The commercial devices mentioned by the reviewer may be based on the piezoelectric effect and capacitive effect, rather than the capacitive piezotronic effect. The relationship between them has been illustrated as follows:

Firstly, we would like to emphasize the definition of the capacitive piezotronic effect in our work (line 136-139): *“Thus, the strain-induced piezoelectric polarization can effectively modulate the **interface width**, so as to control the **junction capacitance under AC settings**. This is the capacitive piezotronic effect, which differs from the previous resistive piezotronic effect that uses piezopotential to control the charge carrier transport of DC electronic devices.”* The capacitive piezotronic effect we proposed here is an **interfacing** modulation effect coupling the **piezoelectricity and semiconductor properties**.

The reviewer introduced the PCB3801 accelerometers as a “capacitive piezotronics” based device. Unfortunately, the detailed information of PCB3801 accelerometers mentioned by the reviewer is no longer accessible given its age. But we believe that its basic mechanism may be similar as the new devices made by “PCB Piezotronics”. Actually, we have already known about the PCB Piezotronics Co. Ltd. and their “capacitive” accelerometer devices. However, these “capacitive” accelerometers are not based on the “capacitive piezotronics”. According to the PCB Piezotronics website, their accelerometers are mostly ICP or MEMS accelerometers. For ICP devices, “under acceleration, the seismic mass causes stress on the sensing crystals which results in a **proportional electrical output**.” And for MEMS devices, “this configuration forms two **air gap capacitors** between the proof mass and upper and lower plates.” Thus, we could summarize that the PCB Piezotronics accelerometers are actually based on the piezoelectric effect or capacitive effect (by tuning the thickness of dielectric layer or **bulk** dielectric properties), rather than an **interfacial** modulation in the “capacitive piezotronics”. On the contrary, a typical capacitive piezotronic device aims at regulating the **interface width** of the heterostructure, so as to change the **junction capacitance**. Therefore, we believe that the capacitive piezotronic devices are different from the commercial devices mentioned by the reviewer.

(3) Although these early devices were based on different materials, it is worth mentioning this historical context to clarify the relationship of this work to earlier ones.

Thanks for the reviewer’s suggestion. We think that the relationship between our work and the

earlier ones should be claimed earlier and more explicitly, and a more detailed distinction among all of the common force or acceleration sensing mechanisms is also needed. Thus, we added **Supplementary Note 7** to clarify the piezoelectric effect, capacitive effect and resistive effect based devices. Meanwhile, we reorganized the original “Relationship between capacitive piezotronics, resistive piezotronics and traditional capacitive effect” section, which was marked in red, and moved it to follow the mechanism section.

We are grateful for the reviewer’s suggestions and guidance for our work and we hope our response would address the reviewer’s question.

Question #1.3

From the very beginning of the manuscript, it should be clearly stated that the study is devoted to piezoelectric semiconductors. In the current version, this fundamental definition appears only in line 122.

Response #1.3:

Thanks for the reviewer’s valuable suggestions and we agree that the materials—piezoelectric semiconductor—used in our work should be clearly clarified at the beginning of the manuscript. We have revised the related parts in the manuscript.

- In line 98-100 in the main text: “*Here, we report the first study of capacitive piezotronics, demonstrating that polarization potential in **piezoelectric semiconductor** can be used as a “gate” signal to reversibly tune the interface width of heterostructures, so as to control the junction capacitance under high frequency AC settings...*”.

Here we would also clarify that we actually have mentioned these statements before line 122 for several times in our original manuscript. The related descriptions are listed as follows:

- At the fourth sentence in abstract of the original manuscript: “*...Here we report the capacitive piezotronics, a distinctive regulation mechanism that is widely present in **piezoelectric semiconductors** (ZnO, GaN, etc.)...*”. This sentence have been revised in our new manuscript as “*...It is a distinctive regulation mechanism that is widely present in **piezoelectric semiconductors** (ZnO, GaN, etc.)...*” (line 43-44)
- At the last sentence in abstract of the original manuscript: “*...This study gives a brand-new strain-tuned AC electronics based on **piezoelectric semiconductors**...*”

We thank the reviewers for these suggestions. And we hope that these revisions could help the readers to understand the scope and context of our work from the very beginning.

Question #1.4

Care must be taken to ensure that all variables described in the descriptive sections of the text (voltage, for instance) are consistently and accurately related to the precise physical quantities and notations used in the formulas and figures.

Response #1.4:

We sincerely thank the reviewer for this valuable suggestion. In response, we have conducted a thorough, line-by-line check of the entire manuscript and supplementary information to ensure all variables are perfectly aligned with their representations in formulas and figures. The main revisions include:

- (1) The applied direct and alternating voltage have been all denoted as “ V_{DC} ” and “ V_{AC} ”, respectively, in **Figure 1**. Also, revisions include:
 - In line 113-115 in section “Mechanism of the capacitive piezotronics”: “...*Under AC settings, the **applied alternating voltage** (V_{AC}) results in the periodic migration of free carriers at the interface, thereby inducing a fluctuation in the amount of charged atoms within the depletion region...*”
 - In line 212-214 in section “Interface width engineering in single Schottky junction via piezoelectric polarization”: “...*the electrical measurement system that uses a DC voltage to apply a bias to the single Schottky junction and simultaneously uses a 2 kHz AC signal (V_{AC}) to conduct the capacitance measurements...*”
 - In line 415-416 in “Methods”: “...*a small V_{AC} with an amplitude of 200 mV and a frequency ranging from 1 to 10 kHz was applied to the capacitive piezotronic devices...*”
- (2) The bias applied in measurements have been all denoted as “(DC)Bias” in **Figure 3, Figure 4, Supplementary Figures 2, 5, 7, 8, 14, 17-27, 29, 30**. Also, revisions include:
 - In line 193 in Supplementary Note 3.2: “...*It is due to the nonlinear dependence between the junction capacitance and the **bias**...*”
 - In line 318-320 in Supplementary Note 6.1: “...*Note that the junction L is forward-biased under positive **bias**, while the junction R is forward-biased under negative **bias**...*”
 - In line 333-335 in Supplementary Note 6.2: “...*the junction L is forward-biased at positive **bias** and thus the corresponding excess capacitance occurs at positive **bias**; while the interface traps located at junction R induces the excess capacitance at negative **bias**...*”
- (3) The output signals in frequency modulation or filtering circuit have been all denoted as “Output signal” in **Figure 5** and **Supplementary Figures 9-12**.

(4) The elementary charge has been denoted as “ q ” in **Figure 2** and **Supplementary Figure 17**, in formulas (3), (4), (S-18) to (S-22), (S-26) to (S-33), (S-35) to (S-36) and (S-38) to (S-39). Also, revisions include:

- In line 194 in section “Theoretical simulations of capacitive piezotronics effect”: “...where q is the elementary charge...”
- In line 464 in the caption of Figure 2b: “... q represents the elementary charge...”
- In line 140 in Supplementary Note 2.2: “... q is the elementary charge...”

(5) The interface barrier height has been all denoted as “ φ ” in **Figure 1**, in formula (S-23), (S-26), (S-32) and (S-33). Also, revisions include:

- In line 457-458 in the caption of Figure 1b: “... W_0 is the initial interface width of Schottky junction, $\Delta\varphi$ and ΔW represent the change of the **interface barrier height** and width, respectively...”
- In line 144-145 in Supplementary Note 2.2: “... φ_1 and φ_2 are the **interface barrier heights** of the Schottky junctions L and R ...”
- In line 157-158 in Supplementary Note 3.1: “...the **interface barrier height** (φ) and width (W) are closely related to it...”
- In line 232-234 in Supplementary Note 4.2: “...which also represents the **interface barrier heights are equal here** ($\varphi_1 = \varphi_2$). Specifically, when the configuration of dual Schottky junctions is perfectly symmetric ($\varphi_1 = \varphi_2$ at zero bias)...”
- In line 244 in Supplementary Note 4.2: “...and thus $V_{bi1} > V_{bi2}$, $\varphi_1 > \varphi_2$...”
- In line 301-302 in Supplementary Note 5: “...when the configuration of dual Schottky junctions is perfectly symmetric ($\varphi_1 = \varphi_2$ at zero bias)...”
- In line 572-573 in the caption of Supplementary Figure 20b, c: “...The asymmetry in dual Schottky junctions is set by the differences between **interface barrier heights** ($\Delta BH = \varphi_2 - \varphi_1$).”
- In line 595 in the caption of Supplementary Figure 23a, b: “...($\Delta BH = \varphi_2 - \varphi_1$)...”

(6) The ratio of the junction resistances has been denoted as “ r ” in formula (S-15), (S-16), (S-22), (S-24), (S-25). Also, revisions include:

- In line 120 in Supplementary Note 2.2: “...where r is the **ratio of the junction resistances** R_2 and R_1 ...”
- In line 204-205 in Supplementary Note 4.1: “...the low-frequency C-V characteristics exhibit an

evident dependence on the *ratio of junction resistances* (r)...”

Other detailed revisions include:

- The word “*Resistor*” has been replaced by “*Resistance*” in **Figure 1a**.
- The formula of resistance and capacitance have been replaced by “ $R \propto \exp\left(\frac{q\phi}{kT}\right)$ ” and “ $C \propto \frac{1}{W}$ ” in **Figure 1a, b**.
- The term “*junction width*” has been replaced by “*interface width*” in **Figure 1b and 4a**.
- The notations in equivalent circuit in **Figure 4b**, “ C_1 ” “ R_1 ” “ C_2 ” “ R_2 ”, have been replaced by “ C_L ” “ R_L ” “ C_R ” “ R_R ”, respectively.
- The change in built-in electric potential induced by piezoelectric polarization, “ V_{piezo} ”, has been replaced by “ ΔV_{bi} ” and the built-in potential after being regulated by polarization, “ V_{bi} ”, has been replaced by “ V_{bi} ” in **Supplementary Figure 18**.

The manuscript has been significantly improved by this process, and we are grateful for the reviewer’s suggestions and guidance for our work.

Question #2

The second key questions are devoted to presenting description of experimental results and experiment.

Response #2:

Thanks for the reviewer’s constructive comment. A clear description of experimental results and experiment is indeed crucial for understanding our work. We have made comprehensive revisions to the manuscript and supplementary materials in response to your subsequent Questions #2.1-2.5.

Question #2.1

I believe the manuscript lacks a clear physical representation of the device under study. In addition to schematic diagrams, it would be helpful to include a more specific illustration or description demonstrating realistic pressure application to the device. Furthermore, I would appreciate a photograph or visual representation clearly demonstrating:

Supplementary, Line 49: “External forces were applied via a high-resolution translation stage, with the force-displacement relationship quantitatively monitored in situ using a calibrated pressure sensor”

Response #2.1:

Thank you for this constructive suggestion. Here we have provided **Figure R1** to show the optical

images of the single and dual Schottky junctions. And a photograph demonstrating realistic pressure application to the device is included in **Figure R2**, where “*External forces were applied via a high-resolution translation stage, with the force-displacement relationship quantitatively monitored in situ using a calibrated pressure sensor*”.

In revisions, these figures have been added into **Supplementary Figure 2** and **Supplementary Figure 3** to further improve our work.

Figure R1 | Capacitive piezotronic devices. **a**, The device based on dual Schottky junctions. **b**, The device based on single Schottky junction. **c**, The optical image of the Schottky contact (Pt/*n*-GaN). Force is applied on the circular area and the lead wires are connected to the strip area. The interface area $S = 0.8296 \text{ mm}^2$. **d**, The optical image of the Ohmic contact (Ti/Al/Ni/Au/*n*-GaN).

Figure R2 | Measuring system. **a**, A total view of the measuring system. **b**, Force application and sensing module, containing an indenter, a sensor and a translation stage. **c**, The indenter. **d**, The software for data collection and processing.

Question #2.2

The supplementary materials contain numerous virtually identical experimental curves (capacitance versus voltage), making them difficult to understand. Furthermore, frequent references to 20 figures in the main text significantly complicate reading and disrupt the flow. I propose reorganizing the article so that the main text remains self-contained and easily understandable. Instead of citing supplementary materials multiple times on each page of the main article, it would be more effective to structure the supplementary materials so as to clearly indicate which sections or figures provide additional explanations or confirm specific parts of the main text.

Response #2.2:

Thank the reviewer's suggestion. We have reorganized both of the main text and the supplementary information.

(1) The detailed descriptions on device fabrication, experimental procedures and physical mechanisms are now added into the main text to make it self-contained. Nevertheless, due to space constraints, we still have to use some references to the supplementary information in the main text. We have endeavored to ensure that readers can grasp the core meaning of the main text without needing to consult the supplementary information. Revisions include:

- In line 248-252 in section “Capacitive piezotronics in dual Schottky junctions”: “...*The electrical measurement system in this experiment utilizes a DC voltage to apply bias between junction L and R, which is respectively reverse-biased (L) and forward-biased (R) at negative DC voltage, while overturned at positive DC voltage, and simultaneously utilizes a V_{AC} signal to conduct capacitance measurements under such bias conditions...*”
- In line 271-276 in section “Capacitive piezotronics in dual Schottky junctions”: “...*Furthermore, because the valley point occurs when the dual junctions are balanced, the valley capacitance could reflect the variation of each junction. Accordingly, we can obtain the changes of interface width of dual Schottky junctions from the change in the valley capacitance (Supplementary Note 4), which respectively reaches 5.1 nm (Fig. 4e) and 0.6 nm (Fig. 4h) at 2.42 MPa. The smaller variation in the interface width with force on junction R could be attributed to the higher initial doping concentration near junction R...*”
- In line 280-315 in section “Capacitive piezotronics in dual Schottky junctions”:
“*The force-dependent variation in C-V curves arises from the asymmetric modulation on dual Schottky junctions via the capacitive piezotronic effect. In the absence of an external loading force, both the band structure and carrier distribution of the dual Schottky junctions remain symmetric, resulting in C-V curves centered at zero bias. When a loading force is applied to only one of the Schottky junctions, the strain-induced negative piezoelectric polarization charges*”

increase both of the built-in electric potential and the interface width of the stressed junction. This asymmetric stimulation requires an additional bias to compensate for the difference in built-in electric potentials, resulting in an obvious shift of the C-V curves. Under low frequencies, taking junction L under loading force as an example, the entire C-V curve shifts toward the positive direction to compensate for the asymmetric potential (Fig. 4c). Consequently, the peak in the negative bias region shifts closer to zero bias. Note that the peak capacitance of C-V curve in low-frequency is primarily dominated by the interface width of the reverse-biased junction (junction L in the negative voltage region and junction R in the positive region). Thus, the peak shifting closer to zero bias is dominated by junction L. An abnormal phenomenon is observed that this peak capacitance increases rather than decreases as the junction L is stressed, which can be attributed to two mechanisms. On the one hand, although the negative polarization charges increase the interface width of the stressed junction, the peak moves to a lower bias owing to the shift of C-V curve, and the lower bias will simultaneously decrease the interface width and hence compensate for the capacitance reduction. On the other hand, the changed effective carrier concentration near the interface also makes a contribution to the increasing capacitance (Supplementary Note 4). For the other peak of C-V curve in the positive bias, the interface width of the unstressed junction R becomes the dominant factor. Since the piezotronic polarization is spatially confined to the stressed junction L, the interface width of the junction R is only influenced by the external bias. As this peak shifts to a higher bias region, the increased bias expands its interface width, and thus decreases the peak capacitance. Conversely, when the force is applied to junction R, the peak value in the negative bias decreases while the other peak value increases (Fig. 4f). This is a unique asymmetric capacitance modulation induced by the capacitive piezotronic effect (Supplementary Figs. 5, 6). Under high-frequency settings, the capacitance of the peak at near-zero bias is dominated by both of the dual Schottky junctions. The interface widths of dual Schottky junctions exhibit opposing variations with one of the junctions stressed: the stressed junction widens, whereas the unstressed junction narrows. The experimental results further suggest that the narrowed junction may play a dominant role (Fig. 4d, g and Supplementary Fig. 7), leading to an increase in peak capacitance under loading forces. Furthermore, this increase is also ascribed to the variation in the effective carrier concentration near the stressed junction (Supplementary Note 5). These results show the strong modulation of capacitive piezotronic effect on dual Schottky junctions, demonstrating significant potential for mechanically controlled high-frequency AC electronics.”

- The section “Methods” has been added to the end of the revised manuscript, which includes detailed device fabrication and experimental procedures.

(2) The supplementary notes have been integrated to fewer parts and rearranged to be more logical and readable:

- The original Supplementary Note 4 has been moved to Supplementary Note 2 in the revised version, and simultaneously more comprehensive descriptions have been added;
- The original Supplementary Note 2 and 3 have been merged into Supplementary Note 3 in the revised version;
- The original Supplementary Note 5 and 6 have been merged into Supplementary Note 4 in the revised version, and simultaneously more comprehensive descriptions have been added;
- The original Supplementary Note 7 has been moved to Supplementary Note 5 in the revised version;
- The original Supplementary Note 8 and 9 have been merged into Supplementary Note 6 in the revised version;
- Supplementary Note 7 has been added into the revised version.

Thanks again for the reviewer's guidance for our work.

Question #2.3

What is the statistics of the results? How many samples were studied?

Response #2.3:

Thanks for the reviewer's comments. We have performed a series of additional experiments, and the measured data of three independent devices with single Schottky junction and three independent devices with dual Schottky junctions are shown in **Figure R3**. The qualitative behaviors remain consistent across all tested devices, indicating that the underlying physical mechanisms is universally valid, which could further confirm our conclusions. Furthermore, the repeatedly results of the devices have been discussed in **Response #2.5**, indicating a highly stable and reproducible modulation behavior. We have added them in **Supplementary Figure 29 and 30**.

Figure R3 | Force-dependent C - V characteristics of multiple independent devices. **a**, Force-dependent C - V characteristics of devices 1-3 based on single Schottky junctions. **b**, Force-dependent low-frequency (1 kHz) C - V characteristics of devices 4-6 based on dual Schottky junctions. **c**, Force-dependent high-frequency (10 kHz) C - V characteristics of devices 4-6 based on dual Schottky junctions.

Question #2.4

The manuscript contains several examples of the use of non-technical or indefinite adjectives:

Supplementary:

- Line 48 ‐a precisely controlled DC bias‐
- Line 50 ‐high-resolution translation stage‐

- Line 145 ...we can get the low-frequency (ω -low) and high-frequency 146 (ω -high) capacitance...
- Line 242 ...as the frequency increases, the excess capacitance response should be weakened because the carrier capture/release rate is too slow to keep up with the high-frequency signals...

Main text:

- Line 95 ...to precisely control the junction capacitance under high frequency AC settings
- Line 174 ...can be precisely controlled by the displacement...
- ...

And more...

It is important that these qualitative terms be accompanied by specific numerical values.

Response #2.4:

Thank you for pointing out these non-technical or indefinite adjectives in our text. Revisions have been made as follows:

(1) Line 48 ...a precisely controlled DC bias...

The revised sentence in line 416-417 in section “Capacitive piezotronic measurements” in “Methods” is: “...*a sweeping DC voltage with a step size of 6 mV was used to apply a bias on the devices...*”

(2) Line 50 ...high-resolution translation stage...

The revised sentence in line 418-419 in section “Capacitive piezotronic measurements” in Methods is: “...*External forces were applied via a high-resolution (10 μ m) translation stage...*”

(3) Line 145 ...we can get the low-frequency (ω -low) and high-frequency 146 (ω -high) capacitance...

Here, we would clarify that the “low-frequency” and “high-frequency” are the limit condition of formula (S-15) when frequency tends to zero (low-frequency limit) and infinity (high-frequency limit), but not an inaccurate qualitative description. To clearly show that, we have made revision in line 126-128 in Supplementary Note 2.2: “...*we can get the equivalent capacitance of dual Schottky junctions at low-frequency limit ($C_{\omega\text{-low}}$, ω tends to zero) and its equivalent capacitance at high-frequency limit ($C_{\omega\text{-high}}$, ω tends to infinity)...*”

(4) Line 242 ...as the frequency increases, the excess capacitance response should be weakened because the carrier capture/release rate is too slow to keep up with the high-frequency signals...

Here, we have estimated the time constant ($\tau \sim 800$ ms) of the carrier capture/release process at the deep-level traps in *n*-GaN, according to (Chattopadhyay, P. *Solid-State Electron.* 1993, 36, 605):

$$\tau = \frac{1}{v_{th}\sigma N_D} \exp\left(\frac{q\psi_s}{kT}\right) \quad (\text{R} - 2)$$

where σ is the capture cross-section of the interface states, v_{th} is the thermal velocity of the carriers, N_D is the carrier concentration, q is the elementary charge, ψ_s is the interface barrier height, k is the Boltzmann constant and T is the temperature.

Obviously, the dynamic behavior with a such large time constant around 800 ms could not keep up with the high-frequency (> 10 kHz) signals.

We have made revision to clarify this issue more clearly in line 320-323 in Supplementary Note 6.1: “...as the frequency increases, the excess capacitance response should be weakened. This is because the time constant of the carrier capture/release process is too large (~ several hundred ms) to keep up with the high-frequency signals (> 10 kHz)...”

(5) Line 95 ...to precisely control the junction capacitance under high frequency AC settings

We would clarify that this sentence is a qualitative summarizing statement, but the word “precisely” is not quite accurate. We have deleted this inaccurate word in the third paragraph of the main text, “...so as to control the junction capacitance under high frequency AC settings...” (line 100-101), and a quantitative description is performed later in the same paragraph: “...It demonstrates a high sensitivity ($\Delta C/p > 110$ fF/mbar in single Schottky junction) in tuning the junction capacitance...” (line 103-104)

(6) Line 174 ...can be precisely controlled by the displacement...

Since the force-displacement relation is nonlinear, its sensitivity cannot be summarized as a constant value. We have measured the force-displacement relation in **Supplementary Figure 3** to provide a further data support.

We have made a revision in line 224-226 in section “Interface width engineering in single Schottky junction via piezoelectric polarization”: “External loading forces were then applied to the Schottky junction along the polarization c -axis of piezoelectric n -GaN, **which were controlled by the displacement of a translation stage (Supplementary Fig. 3)**...”

(7) And more...

- In the Figure 4 caption: “...The C - V characteristics of dual Schottky junctions based device under **1 kHz (c) and 10 kHz (d)** AC settings with forces applied to junction L ...show the C - V characteristics of dual Schottky junctions based device under **1 kHz (f) and 10 kHz (g)** AC settings...” (line 493-496)
- In the Figure 3a caption: “...**2 kHz** V_{AC} and DC bias are simultaneously applied on the electrodes...” (line 476-477)

Question #2.5

Error bars should be discussed for experimental results.

Response #2.5:

Thanks for the reviewer's suggestion. We have performed additional repeated experiments, and added the experimental data with error bars. **Figure R4a-c** and **R5a-c** show the measured $C-V$ curves, and **Figure R4d** and **R5d** show the force-induced changes in capacitance under different bias of single and dual junction devices, respectively. These results show consistent results as our original manuscript, further confirming the validation of the modulation effect and the underlying mechanism. Related description and figures have been added to **Supplementary Figure 30**.

Figure R4 | Repeat $C-V$ measurements on single Schottky junction based device (a-c) and the force-induced change of capacitance under different bias (d).

Figure R5 | Repeat C-V measurements on dual Schottky junction based device (a-c) and the force-induced change of capacitance under different bias (d).

Question #3

The use of the specific words.

Response #3:

Thanks for the reviewer’s constructive comment. We have made comprehensive revisions in response to your subsequent Questions #3.1-3.2.

Question #3.1

Quite often the authors use the words, that in principle may mean what maybe they mean, but in context of the paper they are confusing:

- Line 114 “This dynamic hysteresis phenomenon is fundamentally equivalent to the capacitive reactance (X_C) inherent in MS junction, resembling a charge-discharge process in a conventional capacitor.”

Phenomenon (if consider dynamic hysteresis as a processes can be equivalent to the another process – charging-discharging). But it cannot not be equivalent to the value, such as capacitive reactance (X_C).

- Line 150 “...potential (V_{bi}) of the Schottky junction. It is noted that...” Who noted that?
- Line 177 “...The linear correlation of $(1/C)^2-V$ curve ...” must be changed by: “The linear

(1/C)²-V dependence...”

- Line 195 “...built-in electric potential, which is intuitively demonstrated by the experimental results...”. What means “intuitively” in the context of experimental demonstration?
- Line 335 “...For the capacitive piezotronics, reactance (X) is applied to further describe the effect...”. How the measurable value (X) can be applied to describe?

Response #3.1:

We thank the reviewer for pointing out these confusing words. And, we have addressed each of them with clearer explanations and revisions as follows:

- (1) Line 114 “This dynamic hysteresis phenomenon is fundamentally equivalent to the capacitive reactance (X_C) inherent in MS junction, resembling a charge-discharge process in a conventional capacitor.” Phenomenon (if consider dynamic hysteresis as a processes can be equivalent to the another process – charging-discharging). But it cannot not be equivalent to the value, such as capacitive reactance (X_C).

We acknowledge that our description on the capacitive characteristics of MS junction is confusing here and have provided a revision for clearer explanations in line 112-119 in section “Mechanism of the capacitive piezotronics”: “*Figure 1b illustrates the main principle of the capacitive piezotronics. The amount of charged donors or acceptors within the depletion region can be modulated by an external voltage. Under AC settings, the applied alternating voltage (V_{AC}) results in the periodic migration of free carriers at the interface, thereby inducing a fluctuation in the amount of charged atoms within in the depletion region. **This dynamic behavior resembles a charge-discharge process in a conventional capacitor.** Notably, a charge-discharge process has a typical time constant, finally resulting in a substantial phase shift. Thus, a Schottky junction manifests a measurable capacitive characteristic under AC settings, and this capacitance is directly governed by the amount of charged atoms within the depletion region.*”

- (2) Line 150 “...potential (V_{bi}) of the Schottky junction. It is noted that...” Who noted that?

A proper reference to formula (2) has been added in line 191-192 in section “Theoretical simulations of capacitive piezotronic effect”: “...*It is noted that the interface width is proportional with the square root of V_{bi} at zero reverse bias ($V_R = 0$)²...*” (Sze, S. M. *et al. Physics of Semiconductor Devices*. John Wiley & Sons, 2021)

- (3) Line 177 “...The linear correlation of (1/C)²-V curve...” must be changed by: “The linear (1/C)²-V dependence...”

We thank the reviewer for pointing out our inappropriate usage of “...*The linear correlation of*

$(1/C)^2$ -V curve...” and we have revised it in line 220-221 in section “Interface width engineering in single Schottky junction via piezoelectric polarization”: “...*The **linear** $(1/C)^2$ -V dependence at reverse bias also indicates the successful construction of the single Schottky junction...*”

(4) Line 195 “...built-in electric potential, which is intuitively demonstrated by the experimental results...”. What means “intuitively” in the context of experimental demonstration?

Thank you for pointing that out and we agree that the word “intuitively” is not quite suitable for demonstrating the variation in built-in electric potential. Here we have revised it in line 233-236 in section “Interface width engineering in single Schottky junction via piezoelectric polarization” as follows: “...*Under loading force, the negative piezoelectric polarization charges generated at the interface will repel electrons away from the interface and effectively enhance the built-in electric potential, **which is demonstrated in Figure 3e...***”.

(5) Line 335 “...For the capacitive piezotronics, reactance (X) is applied to further describe the effect...”. How the measurable value (X) can be applied to describe?

We agree that the usage of measurable value (X) is not suitable here and have made revisions on it in line 160-162 in section “Relationship between capacitive piezotronics, resistive piezotronics and traditional capacitive effect”: “...*In contrast, the capacitive piezotronics actually aims at studying the piezotronic modification on semiconductor devices under AC conditions ($\omega > 0$), **where the reactance (X), particularly its capacitive component (C), becomes a critical parameter...***”

The manuscript has been significantly improved by these revisions, and we are grateful for the reviewer’s suggestions and guidance for our work.

Question #3.2

Quite often the adverbs are used instead of adjectives:

- Line 85: “...Particularly, with widely use of...materials...”

And more...

Response #3.2:

We sincerely thank the reviewer for their attention to the grammatical details of our manuscript. We apologize for the improper use of adverbs in the original text and have revised them as follows:

- (1) Line 85: “...Particularly, with widely use of...materials...”

Revision has been made in line 87 in the main text: “...*Particularly, with the **wide** use of sub-10 nm (even sub-5 nm) materials...*”

- (2) Other revisions on the improper usage of adverbs, which is not referred by the reviewer:

- In line 259-261 in section “Capacitive piezotronics in dual Schottky junctions”: “...*It can be seen that the two peaks at low-frequency are slightly different, resulting from the **slight** difference of the two initially fabricated Schottky junctions...*”
- In line 383-384 in section “Summary and outlook”: “...*which enables **mechanical** tuning of communication systems including signal transmission and filtering...*”

We are grateful for the reviewer’s guidance for our work.

Question #4

It is also necessary to carefully review and correct the grammar throughout the manuscript.

Response #4:

We thank the reviewer for this valuable suggestion and apologize for the grammatical errors in our original text. To address this, we have checked the entire manuscript and supplementary materials.

(1) Revisions on the use of punctuations:

In the abstract:

- “...*and substantially improving the filtering characteristics particularly for **high-frequency** noise...*” (line 51-52)
- “...*offering a distinctive approach for **high-quality** communication via mechanical stimuli.*” (line 53-54)

In the main text:

- “...*Particularly, with the wide use of sub-10 nm (even sub-5 nm) materials and development of device miniaturization in **present-day** technology...*” (line 87-88)
- “...*This work provides an in-depth understanding of the discovered capacitive piezotronics towards **next-generation AC electronics**, demonstrating its huge potential in **high-quality** communication.*” (line 107-109)
- “...*and provides a new strategy for developing mechanical-stimulation-controlled devices in **high-frequency communications**.* (line 144-145)

(2) Revisions on use of verbs:

In the main text:

- “*Interfaces of materials and heterostructures **lie** at the heart of the laws in condensed-matter physics...*” (line 60)
- “...*Recently, interface engineering, **that utilizes** piezoelectric³⁻⁶, pyroelectric^{7,8}, ferroelectric⁹,*

or flexoelectric¹⁰⁻¹² effect **to induce** polarization in semiconductors, has inspired emerging fields...” (line 62-63)

- “...Highly sensitive capacitive piezotronics for junction capacitance tuning has been achieved, of which the sensitivity largely **outperforms** that of commercial capacitive pressure sensors...” (line 379-381)
- “...the capacitive piezotronics is widespread in noncentrosymmetric semiconductors including **not only wurtzite structured ZnO and GaN, but also two-dimensional van der Waals materials** (MoS₂, In₂Se₃ etc.) and perovskite materials...” (line 385-387)

In Supplementary Note 6.2:

- “...the investigation into strain-induced band modulation further **provides** a more comprehensive understanding of capacitive piezotronics...” (line 342-343)

(3) Revisions on tense consistency:

In the main text:

- “...which **has also been** the focus of interface engineering by polarization in **recent** years.” (line 76-77)
- “...According to formula (3), the enhancement in built-in electric potential **further leads to** an increase in the interface width, **consequently reducing** the junction capacitance...” (line 236-237)

(4) Revisions on prepositions and prepositional phrases:

In the abstract:

- “...More excitingly, the new effect possesses a capacity of mechanically tuning the transmission signal **in** communication systems with a frequency shift > 11 kHz...” (line 49-51)

In the main text:

- “...providing a possible scheme **for** advanced performance of alternating current (AC) electronics/optoelectronics...” (line 86-87)

(5) Revisions on the use of nouns:

In the main text:

- “...Manipulation of **interfaces** allows emergent effects and functionalities...” (line 61)
- “...which controls the processes of charge carrier separation, **transport**, relaxation or

recombination within the interface region...” (line 83-84)

- “...The equivalent circuit of frequency modulation is schematically shown in Figure 5b, comprising a resonant circuit made of an **inductor** (L) and a capacitive piezotronic device with dual Schottky junctions (serving as a tunable capacitor C')...” (line 350-352)
- “...Thus, our findings open up a new avenue for mechanically tunable AC electronics in semiconductors, which are promising for the new kinds of AC electronic **devices** and electromechanical applications...” (line 388-390)

(6) Revisions on the use of articles:

In the main text:

- “...and substantially increase the cutoff frequency of **a** filtering circuit...” (line 106-107)
- “...This work provides **an** in-depth understanding of the discovered capacitive piezotronics...” (line 107-108)
- “...**a** piezoelectric polarization field can be generated...” (line 127-128)
- “...Figure 3d shows the detailed modulation process in **an** n-GaN based single Schottky junction...” (line 233)

(7) Revisions on other kinds of grammatical errors:

In the second paragraph of main text:

- “...especially **those** regarding communication applications such as high-frequency transport³⁷⁻³⁹ and high-cutoff-frequency radio-frequency diode...” (line 80-81)
- “To verify the above theoretical simulations, we fabricated two-terminal devices **consisting of** a single Schottky junction and an Ohmic contact **on** a piezoelectric n-GaN single crystal with low carrier concentration...” (line 210-212)
- “...Figure 3d shows the **detailed** modulation process in an n-GaN based single Schottky junction...” (line 233)
- “**In practice**, the presence of free carriers in piezoelectric semiconductor will generate an electrostatic screening effect on the produced piezoelectric polarization...” (line 318-319)

We sincerely thank the reviewer for this constructive suggestion again.

In conclusion, I need additional evidence which support the result (see #1 and #2 in my report). It

requires better data presentation and more logic and accuracy in the text. The work demonstrates significant results and deserves publication after significant revision.

Response:

We gratefully thank the reviewer for the comments on our work as “significant results”. We have conducted a thorough and comprehensive revision to address the reviewer’s questions. We hope that our point-to-point responses, combined with the revised text, can provide strong evidence to support our results and can fully address all the reviewer’s concerns.

Reviewer #2 (Remarks to the Author):

General Assessment

The authors present a comprehensive investigation into how strain-induced polarization in piezoelectric semiconductors can be exploited as an interface engineering mechanism to modulate the interface width and the junction capacitance under AC operation in Schottky-diodes structures. The study combines finite element simulations and experimental measurements on single and dual GaN-based Schottky diodes to explore how controlled compressive strain affects the built-in potential, interface width, and capacitance, reporting strain sensitivities exceeding those of commercial capacitive pressure sensors. Furthermore, the authors illustrate the potential of these mechanisms in practical applications by demonstrating implementations in communication circuits such as frequency modulators and filtering systems.

The manuscript provides a substantial amount of material supporting the main claims, including physical descriptions, finite element simulations, and a broad set of experimental results. The mechanisms are interpreted with an appropriate level of detail, and the inclusion of practical applications and comparative analyses strengthens the overall study. Nevertheless, several aspects of the work would benefit from clarification and refinement to further improve its rigor and readability before it can be considered for publication.

Response:

We sincerely thank the reviewer for acknowledging the importance of our work. And we also thank the reviewer's detailed and responsible reviewing of our work.

General Revisions

It should be noted that capacitance variation in piezoelectric and piezotronic devices under mechanical stress has been reported previously; for instance, in bulk ceramics (Cho *et al.*, *JJAP*, 1995 10.1143/JJAP.34.1591) and in ZnO-based Schottky structures (Zhang *et al.*, *Sensors* 2021, 21(6), 2253; <https://doi.org/10.3390/s21062253>). In this respect, the present manuscript goes beyond those studies and offers a conceptual and practical advance.

Nevertheless, in light of the existing literature, the authors are encouraged to cite and briefly discuss these and others prior works, either in the Introduction or main text, and to compare their findings and underlying mechanisms where appropriate. In addition, they are encouraged to balance some novelty statements to ensure that the contribution is presented in a balanced and accurate way. Such refinements would further enhance the credibility of the manuscript and draw clearer attention

to its genuine advances.

Response to General Revision:

Thanks very much for the reviewer's suggestions. We have carefully read the prior works mentioned by the reviewer. These works are related to our study, which propose capacitance modulation methods with piezoelectric effect, but the mechanisms are different from capacitive piezotronics in this work. We have cited these works and made a brief description in the revised manuscript.

Specifically, the definition of the capacitive piezotronic effect in our work is “*Thus, the strain-induced piezoelectric polarization can effectively modulate the **interface width**, so as to control the **junction capacitance** under AC settings.* (line 136-138 in the main text)

Nevertheless, the “capacitance variation in piezoelectric devices” (Cho, Y. *et al. Jpn. J. Appl. Phys.* 1995, 34, 1591) describes another different effect: the force-induced polarization in the traditional piezoelectric device is utilized to induce bound charges on the piezoelectric layer, so as to control its equivalent capacitance. This is actually a traditional piezoelectric device based on a metal-insulator-metal (MIM) structure, while a typical capacitive piezotronic device mainly apply semiconductor heterostructures, such MS junction, MIS junction and *p-n* junction. And for the work reported the “capacitance variation in piezotronic device” (Raoul, J. *et al. Sensors* 2021, 21, 2253), the authors do have found a force-induced variation in *C-V* characteristics of Pt/ZnO/Pt dual Schottky junctions. However, this reported phenomenon was observed only under illumination, and remained many unsolved questions without explanations.

Here, we would like to emphasize our breakthroughs compared to the prior works:

- (1) By introducing the core concept of interface width and emphasizing its importance, a significant progress has been achieved in the theoretical model to explain the polarization-induced variation in junction capacitance; The theoretical model has been further refined in Schottky interface by establishing a correlation between built-in potential and interface width;
- (2) Utilizing Finite Element Method, simulations on the distribution of electric potential and the carrier transport have been achieved, clarifying the inner mechanism of the variation in interface width and junction capacitance under external force;
- (3) Systematic characterizations and detailed analysis in both single Schottky interface and dual Schottky interfaces have been carried out; the changes in built-in potential, interface barrier height and width have all been calculated. Moreover, the universal nature has been proved by conducting same experiments on devices with different materials and structures;
- (4) Comparative experiments in materials with high carrier density and low carrier density have been conducted to further clarify the influence of piezoelectric polarization charges, excluding

the influence of piezoresistive effect, capacitive effect and so on, which have not been done in most of the prior works.

(5) The applications of this effect in frequency modulation and signal filtering have been presented, demonstrating the potential of this effect in communication engineering;

Despite of the discussion above, these prior works do make up some of the background of our works, thus we have added the description about these works into the manuscript and supplementary materials, and also cited them in the references, as well as balanced some novelty statements:

- In the abstract: “...*However, modulating the interface width and its resultant effects under AC settings have not been deeply explored...*” (line 40-41)
- In line 92-95 in the main text: “...*Some preliminary attempts have been made to modulate the interface width using external stimuli, such as mechanical or optical stimuli⁴². However, most of these prior works have not provided systematic experiments and clear mechanism explanations...*” (Raoul, J. *et al. Sensors* 2021, 21, 2253)
- In line 95-97 in the main text: “...*Thus, achieving dynamic manipulation of the interface width without changing the constructed device structure, and exploring the resultant effect are urgently needed, especially for AC electronics/optoelectronics.*” Here we deleted the “...*but it has not been done.*”
- A new Supplementary Note 7 has been added to compare the underlying mechanisms of these prior works and our work.

We sincerely thank the reviewer for this constructive suggestion and we hope that our revisions will improve the quality of our work.

Abstract Revision

The abstract does not clearly specify the type of investigation conducted, whether the reported results are theoretical/simulative or experimental, nor does it explain how the numerical values were obtained. The authors should explicitly state that the study combines finite element simulations and experimental measurements, and that the reported numerical results refer to single and dual GaN-based Schottky devices. This clarification would provide a more accurate and transparent representation of the work.

Response to Abstract Revision:

Thanks for the reviewer’s valuable suggestions. We are sorry that our descriptions are not clear in abstract and we have made the revision as follows (the type of investigation and the source of numerical values have been explicitly clarified by the bold sentences):

“Interface engineering by ionic polarization is vitally important in condensed-matter physics,

*deriving a plethora of distinctive phenomena, such as resistive piezotronics, pyroelectrics, ferroelectrics, flexoelectronics and many more. Most of them focus on the polarization modulated barrier height for effectively controlling carrier transport of DC electronics. However, modulating the interface width and its resultant effects under AC settings have not been deeply explored. Here, we report the capacitive piezotronics, which utilizes the strain-induced piezoelectric polarization to precisely control the interface width of heterostructures and hence modulate the junction capacitance at a high frequency. It is a distinctive regulation mechanism that is widely present in piezoelectric semiconductors (ZnO, GaN, etc.). **Combining theoretical analysis and experiments, our results show that the interface width can be reversibly tuned with amplitudes as high as 10.5 nm in single Schottky junction and 5.1 nm in dual Schottky junctions, respectively, which presents a high strain sensitivity ($\Delta C/p > 110$ fF/mbar in single Schottky junction) and surpasses that of commercial capacitive pressure sensors (~0.1-0.7 fF/mbar). More excitingly, the new effect possesses a capacity of mechanically tuning the transmission signal in communication systems with a frequency shift > 11 kHz, and substantially improving the filtering characteristics particularly for high-frequency noise. This study gives a brand-new strain-tuned AC electronics based on piezoelectric semiconductors, offering a distinctive approach for high-quality communication via mechanical stimuli.***

Thanks again for the reviewer's suggestions and guidance for our work and we hope our response would address the reviewer's question.

Manuscript Revisions

Question #1

In the caption of Figure 1a, please specify that "P" denotes the polarization vector.

Response #1:

Thanks for the reviewer's suggestion and we are sorry that our incomplete annotations hinder the readability of our work. Here we directly replace the "P" by "polarization" in **Figure 1a** to avoid misunderstanding, and attach the revised figure along with the following **Response #2**.

Question #2

In Figure 1b (ii-1, ii-2, ii-3), both compressive and tensile strains are illustrated as acting along the same direction of the c-axis. Since tensile and compressive strain correspond to opposite deformation directions, leading to opposite polarization vectors, showing both with arrows in the same direction is misleading. The schematic should be revised to accurately depict strain directions relative to the

c-axis and the corresponding polarization orientations.

Response #2:

We sincerely thank the reviewer for pointing out the inappropriate usage of the arrows in **Figure 1** and we are sorry about this unclear annotation. Revisions on **Figure 1** are as follows:

Fig. 1 | Mechanism of the capacitive piezotronics.

Question #3

The authors indicate the parameters used in the simulations, but should clarify which

crystallographic polarity (Ga-face or N-face) is considered and under what bias conditions the simulations were performed.

Response #3:

We thank the reviewer's suggestion. Our simulations were performed on Ga-polar (0001) *n*-GaN at zero bias. To further clarify these parameters in manuscript, we have made revisions as follows:

- In line 184-186 in section "Theoretical simulations of capacitive piezotronic effect": "*To demonstrate the principle of the capacitive piezotronic effect, we performed Finite Element Analysis (FEA) on a Schottky junction based on Ga-polar (0001) n-GaN at zero bias, where the intrinsic piezoelectric c-axis of n-GaN is oriented perpendicular to the interface (Method)...*"
- In the section "Theoretical Calculations" in "Methods":
"*The calculation on capacitive piezotronics was performed with the semiconductor module of COMSOL program package through a 3D model of 200 nm wurtzite Ga-polar (0001) n-GaN...*" (line 431-433)
"*...sweeps of strain were conducted at zero bias using stationary solver with a normal physical-control mesh to demonstrate the potential distribution, conduction band energy and carrier concentration distribution...*" (line 441-443)

Question #4

In addition to the schematics, an optical image of both the single and dual Schottky diodes should be provided. The authors should also specify which capacitor area "S" is used in equation (3).

Response #4:

Thanks for the reviewer's suggestions. Here we have provided the optical images of the single and dual Schottky junctions in **Figure R1**, and marked the capacitor area "S" in it. These optical images are all added into **Supplementary Figure 2**, and have been referred in the revised manuscript to further specify the capacitor area "S" in equation:

- In line 210-212 in section "Interface width engineering in single Schottky junction via piezoelectric polarization": "*...we fabricated two-terminal devices consisting of a single Schottky junction and an Ohmic contact on a piezoelectric n-GaN single crystal with low carrier concentration (Supplementary Fig. 2)...*"
- In line 246-248 in section "Capacitive piezotronics in dual Schottky junctions": "*...we further investigated the force-dependent C-V characteristics of dual Schottky junctions by fabricating a two-terminal device based on back-to-back Schottky junctions (Supplementary Fig. 2)...*"

Figure R1 | Capacitive piezotronic devices. **a**, The device based on dual Schottky junctions. **b**, The device based on single Schottky junction. **c**, The optical image of the Schottky contact (Pt/*n*-GaN). Force is applied on the circular area and the lead wires are connected to the strip area. The interface area $S = 0.8296 \text{ mm}^2$. **d**, The optical image of the Ohmic contact (Ti/Al/Ni/Au/*n*-GaN).

Question #5

Since the junction capacitance is analyzed across various bias ranges, including an I - V curve of the Schottky diode under strain-free conditions would provide a more complete picture of the device behavior.

Response #5:

We thank the reviewer for this insightful suggestion. Here, I - V characteristics of the devices under strain-free condition has been provided in **Figure R2**.

The I - V characteristic in **Figure R2a** is based on the device with single Schottky junction fabricated on low-carrier-concentration GaN. It fits well with the current expression in thermionic emission theory:

$$J = \left[A^* T^2 \exp\left(\frac{-q\phi}{kT}\right) \right] \left[\exp\left(\frac{qV}{kT}\right) - 1 \right], \quad (\text{R} - 1)$$

where the J is the current passing through the single Schottky junction, A^* is the effective Richardson constant, T is the temperature, q is the elementary charge, ϕ is the interface barrier height, k is the Boltzmann constant and V is the bias applied to the single Schottky junction. The interface barrier height is $\sim 0.66 \text{ eV}$ obtained by curve fitting.

The I - V characteristics in **Figure R2b, c** are based on the devices with dual Schottky junctions fabricated on low-carrier-concentration GaN and high-carrier-concentration GaN, respectively. Both of them exhibit approximately symmetrical I - V characteristics, indicating the symmetry in device structure. Besides, the current measured in GaN with high carrier concentration is much higher than

that measured in GaN with low carrier concentration, which is owing to the enhanced conductivity in high-carrier-concentration materials.

In revisions, we have added the I - V characteristics into **Supplementary Figure 2**. We hope that these results can strengthen the support for our claims and thank the reviewer for prompting this investigation.

Figure R2 | I - V characteristics of the devices under strain-free condition. **a**, The I - V characteristic of single Schottky junction fabricated on low-carrier-concentration GaN. The experimental results fit well with the theoretical results in thermionic emission theory and the obtained interface barrier height is ~ 0.66 eV. **b**, The I - V characteristic of dual Schottky junctions fabricated on low-carrier-concentration GaN. **c**, The I - V characteristic of dual Schottky junctions fabricated on high-carrier-concentration GaN.

Question #6

The C - V measurements of the single Schottky diode under compressive strain are reported without indicating the frequency of the AC signal. The authors should specify the frequency used, justify their choice, and clarify whether the capacitance variation depends on frequency, as was done for the dual-Schottky configuration. This might be useful to better understand the dual-Schottky junction case.

Response #6:

We thank the reviewer for this valuable and significant suggestion. In **Figure 3b**, a 2 kHz AC signal was applied during C - V measurements of the single Schottky diode under compressive strain. The measured capacitance of a single Schottky junction is theoretically independent on frequency, and similar method with frequencies ranging from 1 kHz to 10 kHz has also been adopted in many other literatures (**Table R1**), which further prove the rationality of our test conditions. Thus, we believe that 2 kHz is a valid choice for our experiments. The reason why the capacitance is independent on frequency is derived as follows:

As shown in **Figure R3**, a single Schottky junction is described as a combination of junction

capacitance (C_j), junction resistance (R_j) and series resistance (R_s). The corresponding impedance (Z_{single}) is:

$$Z_{\text{single}} = \frac{R_j}{1 + i\omega C_j R_j} + R_s, \quad (\text{R} - 2)$$

where ω is the angular frequency of AC signal, and i is the imaginary unit. In our experiments, the frequency of AC signal is 2 kHz, and the single junction capacitance ranges approximately from 0.3 nF to 5 nF. Note that the junction resistance exhibits a strong dependence on the applied bias: generally, in the $\sim\text{M}\Omega$ order under reverse bias while in the $\sim\Omega$ order under forward bias (Korkut, H. *Nano-Micro Lett.* 2013, 5, 34; Anderson, W. A. *et al. Proc. IEEE* 1975, 63, 206; Cohen, L. D. *Proc. IEEE* 1971, 59, 288). The influence of junction resistance cannot be neglected since $(R_j)_{\text{max}} \approx 1/\omega C_j$. However, the influence of series resistance (generally in the $\sim\Omega$ order) can be overlooked under reverse bias due to the fact that it is much smaller than the junction resistance at this bias. Thus, the single Schottky junction under reverse bias can be regarded as a single parallel connection of a resistor R_j and a capacitor C_j . The simplified impedance (Z_{re}) is:

$$Z_{\text{re}} = \frac{R_j}{1 + i\omega C_j R_j}, \quad (\text{R} - 3)$$

However, the influence of series resistance must be taken into account under forward bias, and thus the equivalent circuit of a single Schottky junction under forward bias still maintains a complex structure. The corresponding impedance (Z_{for}) is:

$$Z_{\text{for}} = \frac{R_j}{1 + i\omega C_j R_j} + R_s. \quad (\text{R} - 4)$$

Note that the equivalent parallel circuit model, which regards the entire circuit as a parallel connection of a capacitor (C_{parallel}) and a resistor (R_{parallel}), is applied in our measurements. The measured equivalent capacitance (C_{parallel}) of a single Schottky junction thus satisfies:

$$\frac{1}{Z_{\text{re}}} = i\omega C_{\text{parallel-re}} + \frac{1}{R_{\text{parallel-re}}} \quad (\text{R} - 5)$$

$$\text{or } \frac{1}{Z_{\text{for}}} = i\omega C_{\text{parallel-for}} + \frac{1}{R_{\text{parallel-for}}}. \quad (\text{R} - 6)$$

Therefore, according to formula (R-3) and (R-5), the equivalent capacitance of a single Schottky junction under reverse bias measured by equivalent parallel circuit model is:

$$C_{\text{parallel-re}} = C_j, \quad (\text{R} - 7)$$

and according to formula (R-4) and (R-6), the equivalent capacitance of a single Schottky junction under forward bias measured by equivalent parallel circuit model is:

$$C_{\text{parallel-for}} = \frac{C_j R_j^2}{(R_j + R_s)^2 + \omega^2 C_j^2 R_j^2 R_s^2}. \quad (\text{R} - 8)$$

Obviously, the measured capacitance under reverse bias *via* equivalent parallel circuit model is exactly the single Schottky junction capacitance, showing independence on frequency, junction resistance and series resistance. Results measured under such conditions are quite suitable for studying the influence of capacitive piezotronic effect on interface width and junction capacitance. Therefore, discussion and analysis in our **Figure 3** are all based on the data measured under reverse bias *via* equivalent parallel circuit model.

Furthermore, the above derivation can also be extended to the case of dual Schottky junctions and the corresponding equivalent capacitance has been shown as formula (S-15) in **Supplementary Note 2.2**, clearly showing dependence on frequency. Therefore, it is necessary to investigate the *C-V* characteristics of dual Schottky junction under different AC frequencies, as in **Figure 4**. The detailed derivation of equivalent capacitance of both single Schottky junction and dual Schottky junctions have been added in **Supplementary Note 2**.

In summary, the frequency selection theoretically has no impact on the measured capacitance of the single Schottky junction shown in **Figure 3b**. To further clarify the test conditions, we have made the revision in the section “Capacitive piezotronic measurements” in “Methods”: “...*The equivalent parallel circuit model (C_p) was adopted to obtain the capacitance of these devices...*”

Figure R3 | Equivalent circuit of a single Schottky junction. **a**, Schematic illustration of a single Schottky junction. **b**, Equivalent circuit of a single Schottky junction under forward bias. **c**, Equivalent circuit of a single Schottky junction under reverse bias.

Table R1 | Frequency settings for Schottky junction capacitance measurements

References	Devices	Frequency
Nat. Energy 8 , 504-514 (2023)	CdTe/V-In ₂ S ₃	4 kHz
Energy Environ. Sci. 16 , 4620 (2023)	Pt/ p -Si	10 kHz
Nat. Energy 6 , 63-71 (2021)	FTO/3D(SIG-2D)/spiro-OMeTAD/Au	10 kHz

Nat. Commun. 16 , 8523 (2025)	GDYO/Pt	1-1.732 kHz
Adv. Energy Mater. 9 , 1803243 (2019)	Cs ₂ SnI ₆ /electrolyte	1 kHz
Energy Environ. Sci. 17 , 6234 (2024)	3D Perovskite 3D Perovskite/SIG-I 3D Perovskite/SIG-Br	10 kHz

Question #7

Supplementary Figure 3 should be corrected, as it currently shows the same strain direction for both compressive and tensile loading.

Response #7:

Thank the reviewer for the suggestion. We are sorry for the annotation mistake in the origin **Supplementary Figure 3 (Supplementary Figure 16 after revision)** and have revised the figure in supplementary materials.

Supplementary Figure 3 (Supplementary Figure 16 after revision) | Modulation mechanism of piezoelectric polarization on the built-in potential of Schottky junction at compressive strain (a), strain free (b) and tensile strain (c) conditions.

Question #8

In the section “Capacitive piezotronics in dual Schottky junctions”, the authors should specify on which junction the bias is applied and explain why they expect that loading one junction does not affect the interface width of the other (line 210-212). They should add a reference to this sentence.

Response #8:

Thanks for the reviewer’s constructive suggestions.

(1) Which junction the bias is applied?

We would like to clarify that, during the C-V characterizations of dual Schottky junctions, the

DC voltage aims at applying bias between junction L and R rather than separately to one of the junctions. Owing to the symmetric structure of this device, the applied voltage between the two junctions leads to the different bias conditions on each junction. For example, under a negative voltage, junction L is reverse-biased while junction R is forward-biased. For clarity, revisions have been added in line 248-252 in section “Capacitive piezotronics in dual Schottky junctions”: “...*The electrical measurement system in this experiment utilizes a DC voltage to apply bias between junction L and R, which is respectively reverse-biased (L) and forward-biased (R) at negative DC voltage, while overturned at positive DC voltage, and simultaneously utilizes a V_{AC} signal to conduct capacitance measurements under such bias conditions...*”

(2) Why they expect that loading one junction does not affect the interface width of the other (line 210-212).

We are sorry for the misunderstanding. This misread is because our previous expression “...*has no influence on the other junction...*” was ambiguous. Our real meaning is that the external force and the piezoelectric polarization is spatially localized only in the stressed junction (but of course, the unstressed junction is also influenced due to voltage division). This conclusion is further confirmed by a finite element simulation (**Figure R4**). And, it has been also reflected in literatures which use piezotronic effect to realize pressure imaging (Liu, S. *et al. Adv. Mater.*, 2019, 31, 1905436). We have revised this sentence in line 253-254 in section “Capacitive piezotronics in dual Schottky junctions”: “...*The piezotronic polarization is spatially confined to the stressed junction...*”

We hope that this revision has resolved the ambiguities, and we thank the reviewer’s help in improving the quality of our work.

Figure R4 | Analysis of the deformation of two-terminal devices under loading force. a, Deformation of the two-terminal device with 2 MPa pressure applied on one of the junctions. The deformed area is concentrated within a range of ~1 mm near the stressed junction. **b,** The

deformation distribution along x -axis ($y = 5$ mm) at the top surface of GaN. Area near the unstressed junction (x ranges from 7 to 8 mm) has almost no change. **c**, The deformation distribution along y -axis ($x = 2.5$ mm) at the top surface of GaN.

Question #9

To improve readability, the color scheme of Figure 4b should differ from the subsequent figures. In reference to the sentence at line 217 the authors could also specify the parameters (junction area,..) that make the two Schottky junctions different and explain how these differences lead to asymmetric C - V peaks. The two Schottky barriers, indeed, are considered symmetrical over the manuscript. Additionally, there appears to be an inconsistency: in Figure 4b the left peak is higher than the right one, while the opposite occurs in subsequent figures.

Response #9:

Thank the reviewer for the suggestions. We have changed the color scheme in **Figure 4b** to distinguish it clearly from subsequent figures (**Figure. 4c–h**). The revised **Figure 4** has been provided in the following **Response #10**.

Actually, the two Schottky junctions are designed to be symmetric in geometry with the same materials (Pt/ n -GaN), same junction areas (as illustrated in **Response #4**), same preparation processes, *etc.*. However, due to the limitation of preparation processes, the GaN layer in our device is not perfectly uniform, and the MS contacts are not ideally symmetric.

The asymmetric C - V peaks observed without loading force could be attributed to the relative higher initial doping concentration near the junction R. A higher doping concentration leads to a narrower interface width and thus a higher capacitance. This hypothesis is further confirmed by the difference in the modulation performance with each junction stressed, which is further discussed in **Response #12**. And we have carefully checked our original data of **Figure 4b**. We found that we have made a mistake that the x -axis was labeled into the opposite direction. We have made revision in the manuscript, and we sincerely thank again for the reviewer's careful observation, and apologize for the confusion caused by the mistake.

Question #10

Figures 4c and 4f are not self-explanatory. It is unclear which curve corresponds to the strain-free case. The referee suggests replacing them with Supplementary Figures 8a and 8c, which are clearer. The authors should also indicate, at least in the caption, the low and high frequencies used for the strain-dependent measurements. Moreover, the rightward shift of the C - V curve under strain is evident near the valley point; the arrows should be repositioned accordingly.

Response #10:

Thanks for the reviewer's suggestion for improving the presentation of our work. We have made revisions in **Figure 4c, f** referring to the style of the original **Supplementary Figure 8a, c (Supplementary Figure 21 after revision)**. And, the arrow has been moved to the valley point to present the evident shift of *C-V* curve, making the curves under different pressure conditions easier to distinguish. The revised **Figure 4** has been provided at the end of this response.

The low and high frequencies of AC signals used in measuring *C-V* curves of dual Schottky junctions are 1 kHz and 10 kHz, respectively. Revisions have been made in the caption of **Figure 4**:

- “...*The C-V characteristics of dual Schottky junctions based device under 1 kHz (c) and 10 kHz (d) AC settings with forces applied to junction L...*” (line 493-494)
- “...*Respectively, f, g show the C-V characteristics of dual Schottky junctions based device under 1 kHz (f) and 10 kHz (g) AC settings, with forces applied to junction R...*” (line 495-496)

We sincerely thank the reviewer again for prompting these important advices, which has significantly improved our work.

Fig. 4 | Capacitive piezotronics in dual Schottky junctions. **a**, Schematics of dual Schottky junctions and mechanism of piezoelectric polarization to modify the interface width in dual Schottky junctions. W_0 represents the initial interface width without force (68 nm at zero bias) and ΔW represents the change of interface width under loading force. **b**, Dispersive behavior of $C-V$ characteristics in dual Schottky junctions. The inset presents the equivalent circuit of the dual Schottky junctions. **c, d**, The $C-V$ characteristics of dual Schottky junctions based device under **1 kHz (c) and 10 kHz (d)** AC settings with forces applied to junction L. **e**, The variation of interface width versus pressure corresponding to (c). Respectively, **f, g** show the $C-V$ characteristics of dual Schottky junctions based device under **1 kHz (f) and 10 kHz (g)** AC settings, with forces applied to junction R. **h**, The variation of interface width versus pressure corresponding to (f).

Question #11

Figure 4c shows that at low bias the C - V curve valley shifts to the right when the left junction is under strain. What do they associate with the leftward shift observed at higher bias?

Response #11:

We are grateful for the reviewer's attention to the curve shift, which is actually a core phenomenon and key evidence for the capacitive piezotronics. In **Figure 4c**, the whole C - V curve indeed shifts to the right at low bias. The "leftward shift" of the C - V curve at higher bias is in fact a visual illusion induced by the variation of the peak values, that the left peak value increases and the right peak value decreases. In fact, the curve at higher bias do not shift to left. A detailed explanation for the variation in value peak has been provided in **Response #13-14**. We hope our response can address your question.

Question #12

Just for clarity, the authors could specify why a smaller capacitance variation is observed when the right junction is loaded at low frequency, in Figure 4f.

Response #12:

Thank the reviewer for this suggestion. The smaller capacitance variation observed when the right junction is loaded could be attributed to the higher initial doping concentration near the right junction, which is an intrinsic problem in the GaN we purchased.

As shown in **Figure 4**, the right peak (dominated by the junction R) is initially a little higher than the left peak in the C - V curves without loading force. This phenomenon indicates a higher doping carrier concentration near the junction R. Note that the interface widths of junction L (W_1) and junction R (W_2) at zero bias are:

$$W_1 = \sqrt{\frac{2\varepsilon_S V_{bi1}}{qN_1}}, \quad (\text{R} - 9)$$

$$W_2 = \sqrt{\frac{2\varepsilon_S V_{bi2}}{qN_2}}, \quad (\text{R} - 10)$$

where ε_S is the dielectric constant of the semiconductor, V_{bi1} and V_{bi2} are the built-in electric potential of junction L and junction R, respectively, N_1 and N_2 are the doping concentration of junction L and R, respectively. Since $N_2 > N_1$, the polarization-induced interface width increment in junction R would be smaller under the same loading force. This leads to a smaller capacitance variation when the right junction is loaded at low frequency.

We have added this explanation in line 275-276 in section “Capacitive piezotronics in dual Schottky junctions”: “...*The smaller variation in the interface width with force on junction R could be attributed to the higher initial doping concentration near junction R...*”

Question #13

Supplementary Figure 9 does not seem consistent with the description provided in the manuscript. Why is the barrier height of the left junction in reverse mode and under strain (i) lower than in the strain-free condition? Why is the depletion width in that case smaller than in the strain-free configuration? When the strain is applied to the right junction, why is the left-junction barrier height higher than in the other two cases?

One would expect the strained junction in reverse mode to exhibit a higher barrier and wider depletion region. If these plots represent a different potential distribution rather than local barrier heights, the authors should clarify this explicitly. In that case, the statement around lines 210–212 regarding the independence of each junction under strain may need reconsideration.

Response #13:

Thanks for the reviewer’s insightful comment. We are sorry that our explanations are not clear enough. The original **Supplementary Figure 9 (Supplementary Figure 5 after revision)** shows the interface widths of the dual junctions **at the peak position**, while the peak position does not remain steadily but shifts under external force. Thus, the core mechanism for these questions is actually the same: the interface width is not only modulated by the piezoelectric polarization charges, but also the external bias where the peak occurs. We have revised the description and made revisions in the manuscript.

(1) Why is the barrier height of the left junction in reverse mode and under strain (i) lower than in the strain-free condition? Why is the depletion width in that case smaller than in the strain-free configuration?

Here the interface barrier height and width are actually coupled and have the same variation behavior, thus we take the interface width as an example for the following explanation:

For junction L, its interface width is modulated by both of the piezoelectric polarization charges and the external bias: a higher negative bias would increase its interface width, while a lower negative bias would decrease it. Notably, the *C-V* curve would shift towards positive bias when the junction L is stressed, thus the peak in the negative bias region shifts closer to zero bias. As a result, at this peak position, the lower negative bias decreases the interface width of junction L as well as its interface barrier height, compensating for the increase induced by the polarization charges.

Revisions have been made in the section “Capacitive piezotronics in dual Schottky junctions”:
“...An abnormal phenomenon is observed that this peak capacitance increases rather than decreases as the junction L is stressed, which can be attributed to two mechanisms. On the one hand, although the negative polarization charges increase the interface width of the stressed junction, the peak moves to a lower bias owing to the shift of C-V curve, and the lower bias will simultaneously decrease the interface width and hence compensate for the capacitance reduction...” (line 293-297)

Actually, the real interface barrier height was not shown in **Supplementary Figure 9**. We are sorry for the misunderstanding, and we have revised it in the revised Supplementary information (**Supplementary Figure 5 after revision**) to avoid possible misunderstanding.

(2) When the strain is applied to the right junction, why is the left-junction barrier height higher than in the other two cases?

As mentioned above, the interface barrier height is not shown in **Supplementary Figure 9** (**Supplementary Figure 5 after revision**), but have the same variation behavior as the interface width. Thus, we will also use the interface width for the following explanation:

The interface width is influenced by both the polarization charges and the external bias. Because the piezoelectric polarization is localized at the stressed junction R and the distance between the two junctions is far enough, there are actually no polarization charges distributing near the unstressed junction L (**Response #8** and **Figure R4**). Thus, for the unstressed junction L, the only factor modulating its interface properties is the external bias. Based on the mechanism mentioned above, as junction R is stressed, the peak at negative bias region shifts away from zero bias. Thus, the higher negative bias increases the interface width and barrier height of junction L.

(3) If these plots represent a different potential distribution rather than local barrier heights, the authors should clarify this explicitly.

As we discussed just above, our original **Supplementary Figure 9** (**Supplementary Figure 5 after revision**) actually shows the interface width rather than interface height at the peak position. We are sorry for our unclear description. We have revised the original **Supplementary Figure 9a** (**Supplementary Figure 5a after revision**) and renamed the title of the figure as: “*Interface width of dual Schottky junctions at peak position*”.

(4) In that case, the statement around lines 210–212 regarding the independence of each junction under strain may need reconsideration.

We are sorry that the previous expression “...has no influence on the other junction...” was ambiguous, and we have made revision in section “Capacitive piezotronics in dual Schottky

junctions”: “...*The piezotronic polarization is spatially confined to the stressed junction...*” (line 253-254)

We hope that our response will address your question and improve the readability of our work.

Question #14

The explanation of the asymmetric strain-induced modulation of the C - V peaks needs clarification. One would expect the strained junction in reverse bias to show lower capacitance than the strain-free case, since its depletion region widens. However, the observed peak shows higher capacitance. While the shift of the peak to lower negative biases the authors refer to in line 245-248 compensates for the widened barrier, it does not fully explain the higher capacitance compared to the strain-free situation. A more explicit discussion of the interaction between the two junctions might be needed to explain why the capacitance increases despite the widened barrier.

Response #14:

Thanks for the reviewer’s insightful comment. We sincerely apologize for our unclear explanations for the rising of C - V peak value in the low frequency occasions. Here, the main factors that resulting in the rising of C - V peak are the variation in the external bias and the effective charge concentration. We have provided a clearer description and an explicit numerical simulation to confirm our mechanism.

(1) The effect of the external bias

The peak with an increased capacitance is dominated by the stressed junction. As we stated in **Response #13**, the interface width of the stressed junction is modulated by both of the piezoelectric polarization charges and the external bias. Although the negative polarization charges increase its interface width, the shift of C - V curve moves the peak to a lower bias region and the lower bias simultaneously decreases the interface width, thereby making a compensation for the reduced capacitance.

We have revised the third paragraph of section “Capacitive piezotronics in dual Schottky junctions”: “...*Note that the peak capacitance of C - V curve in low-frequency is primarily dominated by the interface width of the reverse-biased junction (junction L in the negative voltage region and junction R in the positive region). Thus, the peak shifting closer to zero bias is dominated by junction L . An abnormal phenomenon is observed that this peak capacitance increases rather than deceases as the junction L is stressed, which can be attributed to two mechanisms. On the one hand, although the negative polarization charges increase the interface width of the stressed junction, the peak moves to a lower bias owing to the shift of C - V curve, and the lower bias will simultaneously*

decrease the interface width and hence compensate for the capacitance reduction. On the other hand, the changed effective carrier concentration near the interface also makes a contribution to the increasing capacitance (Supplementary Note 4). For the other peak of C-V curve in the positive bias, the interface width of the unstressed junction R becomes the dominant factor. Since the piezotronic polarization is spatially confined to the stressed junction L, the interface width of the junction R is only influenced by the external bias. As this peak shifts to a higher bias region, the increased bias expands its interface width, and thus decreases the peak capacitance. Conversely, when the force is applied to junction R, the peak value in the negative bias decreases while the other peak value increases (Fig. 4f). This is a unique asymmetric capacitance modulation induced by the capacitive piezotronic effect...” (line 290-306)

(2) The effect of the effective charge concentration

Beginning with an ideal model (zeroth-order approximation), we supposed that the change in built-in electric potential (V_{bi}) and interface width (W) perfectly follows a steady relationship:

$$W = \sqrt{\frac{2\varepsilon_S V_{bi}}{qN_d}}, \quad (\text{R} - 11)$$

where V_{bi} is the built-in electric potential and N_d is the doping concentration. This relation is derived from the 1D Poisson's equation:

$$\frac{d^2\psi(x)}{dx^2} = \frac{1}{\varepsilon_S} [qN_d] \quad (\text{R} - 12)$$

where $\psi(x)$ is the electric potential distribution along the x -axis, as shown in **Figure R5a**. Notably, the validation of formula (R-11) requires that the curvature of $\psi(x)$ does not change, so that the change in built-in electric potential and interface width could be perfectly “matched” under an external stress. To meet this requirement, we actually set a hypothesis that the piezoelectric charges distribute normally in a very thin layer and have little influence on the potential curvature. This hypothesis is valid as we explain the single junction model, but not enough in the dual junction model.

The reality is that, as the force increases, the piezoelectric polarization charges change the distribution of free carriers and donors (first order approximation), simultaneously influences the effective charge concentration (N_{eff}):

$$\frac{d^2\psi(x)}{dx^2} = \frac{q}{\varepsilon_S} [N_D + \Delta N] = \frac{q}{\varepsilon_S} N_{eff} \quad (\text{R} - 13)$$

where ΔN is the variation in the concentration. In this way, the curvature of $\psi(x)$ increases and the

band bending becomes much steeper. Consequently, the depletion region essentially “narrows” to accommodate the sharp potential drop, even though the total barrier is larger (**Figure R5a**). This “narrowed” depletion regions results in the “mismatch” between built-in electric potential and interface width, leading to a narrower interface width under same conditions.

Based on this mechanism, we have induced this “mismatch” between built-in electric potential and interface width into our simulations (**Figure R5b**). Results show that the peak shifting closer to zero bias shows an increment, and the other peak shows a reduction, which is consistent with our experiments. This mechanism further explains why the capacitance increases despite the widened barrier.

We have added this analysis in **Supplementary Note 4.3** and referred it in line 297-299 in “Capacitive piezotronics in dual Schottky junctions”: “...On the other hand, the changed effective carrier concentration near the interface also makes a contribution to the increasing capacitance (Supplementary Note 4)...”. And, we hope that our revised explanations and the added supplementary materials have made our description clearer.

Figure R5 | Correction on the theoretical model of dual Schottky junctions. **a**, Under strain free condition, the interface width has a positive relation with built-in electric potential at zero bias (i). As the force increases, zeroth order correction, that the built-in electric potential is enhanced, is applied to represent the influence of polarization charges (i). The first order correction, that the effective carrier concentration is also changed, is applied to further explain the increase of the peak value in low-frequency $C-V$ curves (ii). **b**, The simulated low-frequency $C-V$ curves of dual Schottky junctions under situations of strain free (dashed line), zeroth correction (blue line) and first correction (red line).

Question #15

In Supplementary Note 4, the role of R_s in the theoretical model should be clarified.

Response #15:

Thanks for the reviewer's comment. In fact, the series resistance is regarded as zero in the derivation since its value is too small compared to junction resistance (R_j) at reverse bias:

As is demonstrated in **Response #6**, a single Schottky junction is described as a combination of junction resistance (R_j), junction capacitance (C_j) and series resistance (R_s). Note that junction resistance is in the $\sim M\Omega$ order under reverse bias. On the contrary, the series resistance is generally in the $\sim \Omega$ order (Korkut, H. *Nano-Micro Lett.* 2013, 5, 34; Anderson, W. A. *et al. Proc. IEEE* 1975, 63, 206; Cohen, L. D. *Proc. IEEE* 1971, 59, 288), and hence can be overlooked under reverse bias because it is much smaller. Since our investigation on single Schottky junction are carried out under reverse bias, the influence of series resistance can be neglected in our experiments.

To further prove this, we have provided **Figure R6**, which compares the equivalent capacitances of single Schottky junction with (C_{original}) and without ($C_{\text{simplified}}$) the influence of series resistance. The used parameters are basically consistent with those applied in our experiments. Results show that the two equivalent capacitances tend to be consistent when the junction resistance is much larger than the series resistance. Thus, the neglect of series resistance under this condition is reasonable in our experiments.

Additionally, dual Schottky junctions is composed of two single Schottky junctions (L and R), which are connected in series. Here, the series resistance in the circuit is also overlooked ($R_s = 0 \Omega$), since dual Schottky junctions always exhibits a much higher junction resistance than series resistance whether under forward or reverse bias. On this basis, the impedance and equivalent capacitance of dual Schottky junctions are independent on series resistance.

We have added these detailed descriptions on the role of R_s in line 58-62 in **Supplementary Note 2.1**: "...Note that the junction resistance exhibits a strong dependence on the applied bias: generally, in the $\sim M\Omega$ order under reverse bias while in the $\sim \Omega$ order under forward bias³⁻⁵. The influence of junction resistance cannot be neglected since $(R_j)_{\text{max}} \approx 1/\omega C_j$. However, the influence of series resistance (generally in the $\sim \Omega$ order³⁻⁵) can be overlooked under reverse bias because it is much smaller than the junction resistance..."

Figure R6 | Comparison on equivalent capacitances of single Schottky junction with (C_{original}) and without ($C_{\text{simplified}}$) the influence of series resistance. The two equivalent capacitances tend to be consistent when the junction resistance is much larger than series resistance. So, the series resistance can be neglected under this condition. The parameters used in simulations are basically consistent with those applied in our experiments.

Question #16

In Supplementary Note 5, the derivation of Equation (17) is not clear and a reference should be added.

Response #16:

Thanks for the reviewer’s comment. We have added the full derivation of the Equation (17) in the original supplementary materials (formula (S-37) after revision) (Van Opdorp, C. *et al. Solid-State Electron.* 1967, 10, 401).

In an ideal symmetric structure of dual Schottky junctions, the interface widths of junction L (W_1) and R (W_2) at zero bias satisfy:

$$W_1 = \sqrt{\frac{2\epsilon_S V_{bi1}}{qN_d}}, \tag{R – 14}$$

$$W_2 = \sqrt{\frac{2\epsilon_S V_{bi2}}{qN_d}}, \tag{R – 15}$$

Suppose the junction L is stressed and the junction R is unstressed, the piezoelectric polarization charges will increase the built-in potential of junction L and thus $V_{bi1} > V_{bi2}$. Under this condition, the low-frequency C - V curve of dual Schottky junctions will exhibit a shift in valley, which occurs only when the interface barrier heights of the dual Schottky junctions are equal. The bias where the valley occurs (V_{valley}) thus satisfies:

$$V_{\text{valley}} = \frac{\varphi_1 - \varphi_2}{q} = V_{\text{bi1}} - V_{\text{bi2}}, \quad (\text{R} - 16)$$

where φ_1 and φ_2 are the interface barrier heights of the Schottky junctions L and R. Under this bias condition, the voltage drops over junction L ($V_{1\text{-valley}}$) and R ($V_{2\text{-valley}}$) are equal:

$$\frac{V_{2\text{-valley}}}{V_{1\text{-valley}}} = \frac{R_{2\text{-valley}}}{R_{1\text{-valley}}} = \exp\left(\frac{\varphi_2 - \varphi_1}{kT}\right) \exp\left(\frac{qV_{\text{valley}}}{kT}\right) = 1, \quad (\text{R} - 17)$$

$$V_{1\text{-valley}} = V_{2\text{-valley}} = \frac{V_{\text{valley}}}{2} = \frac{V_{\text{bi1}} - V_{\text{bi2}}}{2}, \quad (\text{R} - 18)$$

where $R_{1\text{-valley}}$ and $R_{2\text{-valley}}$ are the junction resistances of the Schottky junctions L and R at the valley point, k is Boltzmann constant and T is the temperature. In this way, the interface widths of junction L ($W_{1\text{-valley}}$) and R ($W_{2\text{-valley}}$) at valley point are:

$$W_{1\text{-valley}} = \sqrt{\frac{2\varepsilon_S(V_{\text{bi1}} - V_{\text{valley}})}{qN_d}} = \sqrt{\frac{\varepsilon_S(V_{\text{bi1}} + V_{\text{bi2}})}{qN_d}}, \quad (\text{R} - 19)$$

$$W_{2\text{-valley}} = \sqrt{\frac{2\varepsilon_S(V_{\text{bi2}} + V_{\text{valley}})}{qN_d}} = \sqrt{\frac{\varepsilon_S(V_{\text{bi1}} + V_{\text{bi2}})}{qN_d}}. \quad (\text{R} - 20)$$

Here, we define $W_{1\text{-valley}} = W_{2\text{-valley}} = W_{\text{valley}}$. According formulas (R-14), (R-15), (R-19), (R-20), the Equation (17) in original supplementary materials can be derived:

$$\frac{W_1^2 + W_2^2}{2} = W_{\text{valley}}^2. \quad (\text{R} - 21)$$

Similar derivation can be conducted under the case that the junction L is unstressed and the junction R is stressed. We have added these derivations in the revised **Supplementary Note 4** (line 231-258).

Question #17

The clarity of the manuscript could be further improved by adding additional details on the device fabrication, experimental procedures, and physical mechanisms directly within the main text, rather than leaving them mainly in the Supplementary Information.

Response #17:

Thanks for the reviewer's suggestions. We have added the device fabrication, experimental procedures, and physical mechanisms parts into the manuscript:

- In line 248-252 in section: "Capacitive piezotronics in dual Schottky junctions": "...*The electrical measurement system in this experiment utilizes a DC voltage to apply bias between junction L and R, which is respectively reverse-biased (L) and forward-biased (R) at negative DC voltage, while overturned at positive DC voltage, and simultaneously utilizes a V_{AC} signal to conduct capacitance measurements under such bias conditions...*"

- In line 271-276 section “Capacitive piezotronics in dual Schottky junctions”: “...Furthermore, because the valley point occurs when the dual junctions are balanced, the valley capacitance could reflect the variation of each junction. Accordingly, we can obtain the changes of interface width of dual Schottky junctions from the change in the valley capacitance (Supplementary Note 4), which respectively reaches 5.1 nm (Fig. 4e) and 0.6 nm (Fig. 4h) at 2.42 MPa. The smaller variation in the interface width with force on junction R could be attributed to the higher initial doping concentration near junction R...”
- In line 290-313 in section “Capacitive piezotronics in dual Schottky junctions”: “...Note that the peak capacitance of C-V curve in low-frequency is primarily dominated by the interface width of the reverse-biased junction (junction L in the negative voltage region and junction R in the positive region). Thus, the peak shifting closer to zero bias is dominated by junction L. An abnormal phenomenon is observed that this peak capacitance increases rather than decreases as the junction L is stressed, which can be attributed to two mechanisms. On the one hand, although the negative polarization charges increase the interface width of the stressed junction, the peak moves to a lower bias owing to the shift of C-V curve, and the lower bias will simultaneously decrease the interface width and hence compensate for the capacitance reduction. On the other hand, the changed effective carrier concentration near the interface also makes a contribution to the increasing capacitance (Supplementary Note 4). For the other peak of C-V curve in the positive bias, the interface width of the unstressed junction R becomes the dominant factor. Since the piezotronic polarization is spatially confined to the stressed junction L, the interface width of the junction R is only influenced by the external bias. As this peak shifts to a higher bias region, the increased bias expands its interface width, and thus decreases the peak capacitance. Conversely, when the force is applied to junction R, the peak value in the negative bias decreases while the other peak value increases (Fig. 4f). This is a unique asymmetric capacitance modulation induced by the capacitive piezotronic effect (Supplementary Figs. 5, 6). Under high-frequency settings, the capacitance of the peak at near-zero bias is dominated by both of the dual Schottky junctions. The interface widths of dual Schottky junctions exhibit opposing variations with one of the junctions stressed: the stressed junction widens, whereas the unstressed junction narrows. The experimental results further suggest that the narrowed junction may play a dominant role (Fig. 4d, g and Supplementary Fig. 7), leading to an increase in peak capacitance under loading forces. Furthermore, this increase is also ascribed to the variation in the effective carrier concentration near the stressed junction (Supplementary Note 5)....”
- The section “Methods” has been added to the end of the revised manuscript, which includes

detailed device fabrication and experimental procedures.

We are grateful for the reviewer's guidance, which has been invaluable in strengthening this work.

Question #18

The last section of the manuscript "Relationship between capacitive piezotronics, resistive piezotronics and traditional capacitive effect" could be summarized and added in the introduction or in the 'mechanism of the capacitive piezotronics', since the main claims have already been introduced.

Response #18:

Thanks for the reviewer's suggestion. We have reorganized the section of "Relationship between capacitive piezotronics, resistive piezotronics and traditional capacitive effect", and moved this part to follow the mechanism in the revised manuscript and marked this section in red.

Finally, we appreciate the reviewers' suggestions and guidance for our work again. Based on the details discussed above, we have revised the manuscript and the supplementary materials.

Point-to-Point Response to Reviewers' comments:

We are grateful to the reviewers for their constructive and insightful comments on our revised manuscript and supplementary materials. In the pages that follow, we have provided our responses to each of the reviewers' questions

Reviewer #1 (Remarks to the Author):

All comments have been taken into account in the new version, and the quality of the article has been significantly improved. I believe it can be published in its current form.

Response:

Thanks for the reviewer's positive assessment of our work. We are delighted to learn that all of the reviewer's comments have been satisfactorily addressed and we sincerely thank the reviewer for the detailed and responsible reviewing of our work.

Reviewer #2 (Remarks to the Author):

The authors have responded thoroughly to the reviewer's comments. They have referred to prior literature, improved the clarity of several figures, added the requested derivations and theoretical models, and provided additional details in the main text. In particular, the physical interpretation of the C - V curves shape, the observed shift, and the asymmetric capacitance behavior is now clearer and better motivated. Overall, the manuscript has significantly improved. A few points, however, would still benefit from further clarification before publication.

Response:

We sincerely thank the reviewer for the encouraging feedback and for acknowledging that our previous revisions have significantly improved the quality of the manuscript. And we also thank the reviewer's detailed and responsible reviewing of our revised manuscript and supplementary materials. In response to the remaining points, we have provided further clarifications and made comprehensive revisions to the manuscript and supplementary materials.

Abstract revision

While the revised abstract more clearly states that both theoretical analysis and experimental investigations are involved, it remains unclear how the reported quantitative values are obtained and to which specific devices they refer. The authors are encouraged to explicitly state that finite-element

simulations and experimental measurements were carried out on GaN-based (and subsequently ZnO-based) Schottky diodes, and to clarify which results are experimentally measured and which are supported by simulations. Phrases such as “Experimental measurements on GaN-based Schottky diodes show that...” and “Finite-element simulations confirm that...” could be considered.

Moreover, when referring to “high frequency” operation, the authors should provide indicative frequency ranges (e.g., kHz regime) in order to better contextualize the practical applicability of the devices.

Response to Abstract Revision:

Thanks for the reviewer’s valuable suggestions and guidance. In response to these questions, we have made revisions on the abstract and the relevant sections in the main text.

1. While the revised abstract more clearly states that both theoretical analysis and experimental investigations are involved, it remains unclear how the reported quantitative values are obtained and to which specific devices they refer.

We are sorry for the unclear description in our abstract. We would like to clarify that the finite-element analysis in our work mainly aims at revealing and verifying the underlying mechanism of capacitive piezotronics. And, the reported quantitative values, such as the change in interface width and the strain sensitivity, are all based on our experimental results. To further clarify the research methods, we have revised the abstract as follows (bold sentences):

*“Interface engineering by ionic polarization is vitally important in condensed-matter physics, deriving a plethora of distinctive phenomena, such as resistive piezotronics, pyroelectrics, ferroelectrics, flexoelectronics and many more. Most of them focus on the polarization modulated barrier height for effectively controlling carrier transport of DC electronics. However, modulating the interface width and its resultant effects under AC settings have not been deeply explored. Here, we report the capacitive piezotronics, which utilizes the strain-induced piezoelectric polarization to precisely control the interface width of heterostructures and hence modulate the junction capacitance at a high frequency (~kHz-MHz). It is a distinctive regulation mechanism that is widely present in piezoelectric semiconductors (ZnO, GaN, etc.). **This mechanism was rigorously verified by a combination of finite element analysis and experimental measurements in GaN-based Schottky contacts. Our experimental results show that the interface width can be reversibly tuned with amplitudes as high as 10.5 nm in GaN-based single Schottky junction and 5.1 nm in***

GaN-based dual Schottky junctions, which presents a high strain sensitivity ($\Delta C/p > 110$ fF/mbar in single Schottky junction) and surpasses that of commercial capacitive pressure sensors (~ 0.1 - 0.7 fF/mbar). More excitingly, the new effect possesses a capacity of mechanically tuning the transmission signal in communication systems with a frequency shift > 11 kHz, and substantially improving the filtering characteristics particularly for high-frequency noise (> 300 kHz). This study gives a brand-new strain-tuned AC electronics based on piezoelectric semiconductors, offering a distinctive approach for high-quality communication via mechanical stimuli.”

2. Moreover, when referring to “high frequency” operation, the authors should provide indicative frequency ranges (e.g., kHz regime) in order to better contextualize the practical applicability of the devices.

We thank the reviewer for raising this important point. In response, we have carefully reviewed all instances of “high frequency” in the abstract and main text. Regarding those referring to a specific frequency range, we have revised them according to our test conditions:

- In the fourth sentence of the abstract: “...and hence modulate the junction capacitance at a high frequency (\sim kHz-MHz)...”. This frequency range is determined by the test environment in **Figures 3-4** (1-10 kHz) and the signal frequency in **Figure 5** (~ 300 kHz).
- In the eighth sentence of the abstract: “...and substantially improving the filtering characteristics particularly for high-frequency noise (> 300 kHz)...”. This range is determined by the frequency of noise in **Figure 5**.
- In the third paragraph of the main text: “...so as to control the junction capacitance under high frequency AC settings (\sim kHz-MHz)...”
- In the first paragraph of the section “Summary and outlook”: “...in which the interface width and hence the junction capacitance at high-frequency AC settings (\sim kHz-MHz) can be dynamically modulated by strain-induced piezoelectric polarization...”
- In the section “Capacitive piezotronics in dual Schottky junctions”, the high-frequency condition refers to the frequency range ≥ 5 kHz and it has been clarified at a proper position: “...and found two peaks (one around -0.5 V and the other around 0.5 V) at low-frequency (< 5 kHz), while only one peak (around 0 V) at high-frequency (≥ 5 kHz)...”

Note that, in a few instances, the term “high frequency” is used in a more general, introductory

context, where specifying a particular frequency range would be unsuitable. So, we would like to keep them in their original form, such as:

- In the second paragraph of the main text: “...especially those regarding communication applications such as **high-frequency** transport...”
- In the third paragraph of the section “Mechanism of the capacitive piezotronics”: “...especially at **high-frequency** systems...”
-

We hope that these revisions could improve the quality of our work and we sincerely thank the reviewer again for prompting these important advices.

Specific comments to the authors' responses

R1. Although the symbol “P” has been replaced with “polarization” in Fig. 1a, it is still used elsewhere without definition. For consistency, it should be explicitly defined or replaced throughout the manuscript.

Response #1:

Thanks for pointing that out. We have carefully checked the entire manuscript and supplementary materials, and have ensured that the symbol “P” is either explicitly defined or replaced with the full term “polarization”. The revised **Figure 3** and **Supplementary Figure 16** have been provided at the end of this response:

Supplementary Figure 16 | Modulation mechanism of piezoelectric polarization on the built-in potential of Schottky junction at compressive strain (a), strain free (b) and tensile strain (c) conditions.

Figure 3 | Capacitive piezotronics in single Schottky junction.

R6. The authors have provided a convincing theoretical justification for the choice of measurement frequency and for the frequency-independent behavior of a single Schottky junction under reverse bias. While an explicit experimental verification of the absence of frequency dispersion would have been appreciated, the present explanation is sufficient and consistent with established literature.

Response #6:

Thanks for the reviewer's comment. In response, we have provided the C - V characteristics of a single Schottky junction under different frequencies in **Figure R1**. Experimental results show that the capacitance measured at reverse bias remains almost unchanged as the frequency changes, while the capacitance measured at forward bias exhibit a significant change. These results provide a verification of our theoretical justification, and are also consistent with the previous works (such as Supplementary Figure 10 in *Ref.: Georgiadou, D. G. et al. Nat. Electron., 2020, 3, 718-725*).

Figure R1| The C-V characteristics of a single Schottky junction under different frequencies.

R8. The clarification of the measurement setup and the demonstration that strain effects are spatially localized in the stressed junction are appreciated. However, in the low-frequency C–V characteristics of the dual Schottky junction (Figs. 4c and 4f), the reported zero-strain curves exhibit noticeably different peak asymmetries. The authors are encouraged to clarify whether these strain-free curves were acquired under identical measurement and normalization conditions, and whether factors such as mechanical boundary conditions, contact resistance, or low-frequency drift/hysteresis could account for these differences.

Response #8:

We sincerely thank the reviewer for acknowledging our revised manuscript and providing further valuable suggestions.

In response to the reviewer’s questions on **Figure 4c, f**, we have carefully checked our experimental results. We would like to clarify that, these two zero-strain curves were acquired under the same measurement and normalization conditions, and are consistent with each other within experimental precision. To clearly clarify this, we have emphasized these zero-strain curves with the black lines in **Figure R2**, and it can be seen that the two lines exhibit no significant difference. We have revised the caption of **Figure 4**:

“...These results were acquired under identical test conditions and normalization conditions.”

We are sorry for this misunderstanding and we hope that our response would address your

questions.

Figure R2 | The force-dependent C - V characteristics of dual Schottky junctions based device. a, Force on junction L. **b,** Force on junction R. These results are acquired under identical test conditions and normalization conditions. The zero-strain curves are all emphasized in black.

R9. The explanation of the asymmetric C - V peaks in terms of fabrication-related non-idealities (e.g., local variations in effective doping concentration or slightly different metal pad areas) is physically reasonable. The authors are encouraged to state this more explicitly in the main text, for instance by revising the sentence: “It can be seen that the two peaks at low frequency are slightly different, resulting from...” or by referring to this point in the Methods section.

Response #9:

Thanks for the reviewer’s constructive suggestions. As suggested, we have revised the sentence in the second paragraph of the section “Capacitive piezotronics in dual Schottky junctions” to more explicitly state the explanation of the asymmetric C - V peaks:

*“...It can be seen that the two peaks at low-frequency are slightly different, resulting from **the fabrication induced non-idealities of the dual Schottky junctions (e.g., local variations in effective doping concentration or slightly different metal pad areas)**...”*

R14. The revised description of the increase in the capacitance peak is clearer. The authors are encouraged to elaborate slightly more in the main text on the role of the effective charge concentration, while keeping the full quantitative details in the Supplementary Information. This would help convey

a more complete physical picture within the manuscript.

Response #14:

Thanks for the reviewer's insightful comment. To more explicitly demonstrate the role of the effective charge concentration, we have revised the related parts in the third paragraph of the section "Capacitive piezotronics in dual Schottky junctions" as follows:

*"...On the other hand, the changed effective carrier concentration near the interface **enables interfacial electrostatic equilibrium (Fermi Level alignment) to be sustained within a narrower interface width**, which also makes a contribution to the increasing capacitance (Supplementary Note 4)..."*

This revision provides a more detailed qualitative description on the role of the effective carrier concentration and how its variation affects the peak capacitance. We hope that this response could address your question and improve the quality of our work.

Furthermore, although the valley in low-frequency measurements and the peak in high-frequency measurements are experimentally observed at zero bias (Figure 4), statements such as "the valley occurs only when $R_1=R_2$ " (Supplementary Note 4.2) or "the peak locates at zero bias when the junctions are perfectly symmetric" (Supplementary Note 5) may be misleading in the presence of fabrication-induced asymmetries. It would be clearer to specify that the condition $R_1=R_2$ is satisfied at a specific bias point (which may coincide with zero bias), rather than implying perfect structural symmetry.

Response:

Thanks for the constructive suggestion, and we have revised the relevant content as follows:

- In the third paragraph of the section "Capacitive piezotronics in dual Schottky junctions": *"...In the absence of an external loading force, both the low-frequency and high-frequency C-V curves of the dual Schottky junctions are centered at near-zero bias region..."*
- In the first paragraph of Supplementary Note 4.2: *"Based on analysis above, the valley in low-frequency C-V curve occurs only at the bias where the resistances of the dual Schottky junctions are equal ($R_1 = R_2$), which also represents the interface barrier heights are equal here ($\phi_1 = \phi_2$). Generally, this symmetry condition $R_1 = R_2/\phi_1 = \phi_2$ can be satisfied at a specific bias (which may coincide with zero bias if the configuration of dual Schottky junctions is almost symmetric). When*

the force applied on one of the junctions, the induced negative polarization charges will increase its interface barrier height and width and the specific bias that satisfying the symmetry condition will change, with the valley shifts and a variation in C–V curve being induced (Supplementary Fig. 20a)...

- In the first paragraph of Supplementary Note 5: “...*Similar to the case under low-frequency settings, the peak locates at a specific bias (which may coincide with zero bias if the configuration of dual Schottky junctions is almost symmetric) under zero-strain condition, while exhibits a shift when the configuration of dual Schottky junctions is modified by the force-induced piezoelectric polarization potential...*”

Thanks for the reviewer’s guidance again and we hope that our revision could improve our work.

Finally, the authors are encouraged to further refine the language and avoid overly strong expressions such as “abnormal phenomenon”, “huge impact”, or “unexpectedly giant effect”, in order to maintain a balanced scientific tone.

Response:

Thanks for the in-depth reviews to help us improve the manuscript. We have conducted a careful check on our manuscript and supplementary materials and have revised the relevant contents as follows:

- In the third paragraph of the section “Mechanism of the capacitive piezotronics”: “...*which significantly influences the migration and redistribution of carriers at the interface, so as to change the interface width...*”
- In the third paragraph of the main text: “...*A **substantial** capacitive piezotronic effect was discovered in both single and dual Schottky junctions...*”
- In the third paragraph of the section “Capacitive piezotronics in dual Schottky junctions”: “...*An **anomalous** phenomenon is observed that this peak capacitance increases rather than deceases as the junction L is stressed...*”
- In the first paragraph of the section “Summary and outlook”: “...*This effect reveals a **seldom recognized coupling** between piezoelectric polarization and high-frequency AC electronics...*”

Finally, we thank the reviewer for the deep patience and effort in reviewing our manuscript. We are particularly grateful for the detailed logic, linguistic and stylistic suggestions. These insightful

suggestions have significantly refined the presentation of our work and improved the overall readability of the manuscript.

Reviewer #2 (Remarks to the Author):

The authors have adequately addressed all the comments in the revised manuscript. In my opinion, the paper can be accepted for publication in its current version.

Response:

Thanks for the reviewer's positive assessment of our work. We are delighted to learn that all of the reviewer's comments have been adequately addressed and we sincerely thank the reviewer for the detailed and responsible reviewing of our work.

Finally, we sincerely thank the editor and the reviewers for spending so much time and effort on improving the quality of our paper. These constructive and professional suggestions are very helpful for us to upgrade our understandings and investigations and promote us to one step further. Thank you very much!